# Through the Stealth Lens: Attention-Aware Defenses Against Poisoning in RAG

**Sarthak Choudhary** [1]   **Nils Palumbo** [1]   **Ashish Hooda** [1]   **Krishnamurthy (Dj) Dvijotham** [2]   **Somesh Jha** [1]

## Abstract

Retrieval-augmented generation (RAG) systems are vulnerable to attacks that inject poisoned passages into the retrieved context, even at low corruption rates. We show that existing attacks are not designed to be stealthy, allowing reliable detection and mitigation. We formalize a distinguishability-based security game to quantify stealth for such attacks. If a few poisoned passages control the response, they must bias the inference process more than the benign ones, inherently compromising stealth. This motivates analyzing intermediate signals of LLMs, such as attention weights, to approximate the influence of different passages on the response. Leveraging attention weights, we introduce the **Normalized Passage Attention Score** (NPAS) and a lightweight **Attention-Variance Filter** (AV Filter) that flags anomalous passages. Our method improves robustness, yielding up to ∼ **20%** higher accuracy than baseline defenses. We also develop adaptive attacks that attempt to conceal such anomalies, achieving up to **35%** success rate and underscoring the challenges of achieving true stealth in poisoning RAG systems.

## 1. Introduction

Large Language Models (LLMs) have revolutionized various applications with their remarkable generative abilities. However, their reliance on internal knowledge can lead to inaccuracies due to outdated information or hallucinations (Achiam et al., 2023; Brown et al., 2020; Ji et al., 2023). RAG (Guu et al., 2020; Lewis et al., 2020) has emerged as a leading technique to address these limitations by integrating LLMs with external (non-parametric) knowledge retrieved from databases (Borgeaud et al., 2022; Karpukhin et al., 2020). It retrieves a set of relevant passages from a

knowledge database, denoted as the *retrieved set*, and incorporates them into the model's input. This powerful approach underpins critical real-world systems, including Google Search with AI overviews (Google, 2024), WikiChat (Semnani et al., 2023), Bing Search (Microsoft, 2024), Perplexity AI (Perplexity AI, 2024), and LLM agents (Liu, 2022; LangChain, 2024; Shinn et al., 2023; Yao et al., 2023).

The reliance of RAG systems on the retrieved set, however, introduces a significant new security vulnerability: the knowledge database becomes an additional attack surface. Malicious actors can inject harmful content, for example, by manipulating Wikipedia pages, spreading fake news on social media, or hosting malicious websites, to corrupt the information retrieved by the RAG system (Carlini et al., 2024). Consequently, the retrieval of malicious passages by a RAG system, followed by their incorporation into response generation, constitutes a *retrieval corruption attack* (Xiang et al., 2024). Recent instances, such as the PoisonedRAG attack (Zou et al., 2024), demonstrate easily exploitable vulnerabilities: the attacker simply prompts GPT-4 to create the malicious context and inject it into the retrieved set, successfully manipulating the answer by corrupting only a small fraction of the retrieved set (e.g., one out of ten) (Greshake et al., 2023; Zou et al., 2024; Xiang et al., 2024).

Although existing attacks on RAG systems often achieve high success with low corruption rates, they are typically not designed with stealth in mind, leaving them susceptible to detection and mitigation. Ideally, a robust aggregation mechanism would identify inconsistencies between the LLM's output and the dominant (benign) signal in the retrieved set. A significant divergence suggests undue influence from a small, potentially malicious subset of passages. Crucially, to override the benign context, adversarial passages must disproportionately influence the LLM's response. This may necessitate detectable differences from benign passages, leaving behind a malicious trace. The presence of such malicious traces becomes more likely when the adversary cannot compromise the majority of the retrieved set, a particularly challenging task when retrieval is performed over large, diverse corpora like Google Search or Wikipedia (Xiang et al., 2024; Zou et al., 2024; Greshake et al., 2023), or when the retriever is designed to be robust. However, existing attacks largely overlook stealth, relying on weak signals such as perplexity (Jain et al., 2023; Alon & Kam-

---

[1]University of Wisconsin-Madison [2]ServiceNow Research. Correspondence to: Sarthak Choudhary <schoudhary28@wisc.edu>.

*Proceedings of the 43rd International Conference on Machine Learning*, Seoul, South Korea. PMLR 306, 2026. Copyright 2026 by the author(s).

fonas, 2023; Gonen et al., 2022). This raises a fundamental question: *Are existing attacks truly stealthy? If not, can they be detected and mitigated, and how can we develop more sophisticated strategies to enhance their stealth?* We challenge the notion of effortless stealth and define it through a distinguishability security game. We introduce the **Normalized Passage Attention Score (NPAS)**, a metric that aggregates the attention weights assigned to tokens in each passage from the model's response. We demonstrate that existing low-effort attacks leave detectable traces, as adversarial passages attract disproportionately high attention, typically due to phrases containing or strongly implying the adversarial answer.

Our key insight is that under low corruption, any successful attack must concentrate influence in a small subset of retrieved passages; this concentration is detectable via attention-derived influence scores. We operationalize this insight with NPAS and a lightweight **Attention-Variance Filter (AV Filter)** that removes passages with anomalously high normalized attention (see Figure 1 for an overview) [1]. The AV Filter effectively distinguishes malicious passages from benign ones, enabling robust defenses by filtering out potentially malicious passages. To rigorously explore the limits of this defense, we extend jailbreak methodologies to create adaptive attacks that optimize for obscuring attention-based traces and evading the AV Filter, marking progress toward stealthier attacks. Our findings highlight the ongoing arms race between attacks and robust RAG systems by formalizing a security game, demonstrating effective mitigation of existing low-stealth attacks, and revealing the challenges in improving stealth through adaptive attacks. Our contributions are as follows:

- We formalize stealth in RAG poisoning via the Stealth Attack Distinguishability Game (SADG).

- We introduce the Normalized Passage Attention Score (NPAS) to quantify passage-to-response influence and show it becomes skewed under corruption.

- Building on NPAS, we propose the Attention-Variance Filter (AV Filter) and a corresponding defender that reliably distinguishes benign from corrupted retrievals.

- We evaluate across 4 datasets, 5 LLMs, 5 attacks, and develop adaptive attacks that optimize for evading AV Filter, exposing the stealth–effectiveness trade-off.

- We study both white-box settings (access to model internals) and black-box settings using an auxiliary open-source model to compute attention-based signals.

---

[1]Our implementation is available at `https://github.com/sarthak-choudhary/Stealthy_Attacks_Against_RAG`.

## 2. Background and Related Work

**Notations and Definitions.** Table 1 summarizes the key notation used throughout this paper.

*Table 1.* Summary of notation.

| Symbol | Definition |
|---|---|
| $\mathcal{Q}, \mathcal{S}$ | Spaces of queries and responses; $q \in \mathcal{Q}$, $s, s' \in \mathcal{S}$ denote a query, valid response, and adversary's target ($s \neq s'$) |
| $\mathcal{Z}$ | Space of knowledge databases; $z \in \mathcal{Z}$ is a collection of passages $\{z_1, z_2, \ldots, z_n\}$ |
| $z^{(k)}, \mathcal{Z}^{(k)}$ | Subset of $k$ retrieved passages and the space of all such subsets |
| $\Theta$ | Space of RAG architectures; $\theta = (\text{Ret}_\theta, \text{Gen}_\theta, \text{LLM}_\theta)$ comprises retriever, generator, and LLM |

**Retrieval-Augmented Generation.** A RAG pipeline comprises four key components: a knowledge database, a retrieval function, a generation function, and an LLM. The knowledge database consists of a collection of passages sourced from diverse repositories such as Google Search or Wikipedia.

*Step I. Knowledge Retrieval:* The retriever selects the top-$k$ passages relevant to $q$. Formally, $\text{Ret}_\theta : \mathcal{Q} \times \mathcal{Z} \rightarrow \mathcal{Z}^{(k)}$ denotes the retrieval function that returns the top-$k$ passages.

*Step II. Generation*: The generation function utilizes the retrieved set and the LLM, often guided by an instructional prompt $\mathcal{I}$, to produce the final response. Formally, $\text{Gen}_\theta : \mathcal{Q} \times \mathcal{Z}^{(k)} \rightarrow \mathcal{S}$ denotes the generation function that outputs the response $s$.

**Definition 2.1** (RAG System). A Retrieval-Augmented Generation (RAG) system is a function $f_{\text{RAG}} : \mathcal{Q} \times \mathcal{Z} \times \Theta \rightarrow \mathcal{S}$, defined as:

$$f_{\text{RAG}}(q, z, \theta) = \text{Gen}_\theta(q, \text{Ret}_\theta(q, z)) = s.$$

In a standard RAG pipeline (Lewis et al., 2020), the retriever assigns relevance scores to passages independently and selects the top-$k$ passages based on these scores. The retriever's output is:

$$\text{Ret}_\theta(q, z) = z^{(k)} = \{z_{i_1}, z_{i_2}, \ldots, z_{i_k}\}$$

Next, the generation function processes a concatenated sequence consisting of the instructional prompt, the retrieved passages, and the query to produce a response. This is formulated as:

$$\text{Gen}_\theta\left(q, z^{(k)}\right) = \text{LLM}_\theta\left(\text{Concat}(\mathcal{I}, z^{(k)}, q)\right)$$
$$= \text{LLM}_\theta(\mathcal{I} \oplus z_{i_1} \oplus \cdots \oplus z_{i_k} \oplus q),$$

where $\oplus$ denotes the concatenation of text sequences.

**Vulnerabilities in RAG Systems.** An adversary targeting a specific response $s'$ can craft adversarial passages $z_{\text{adv}}$ by

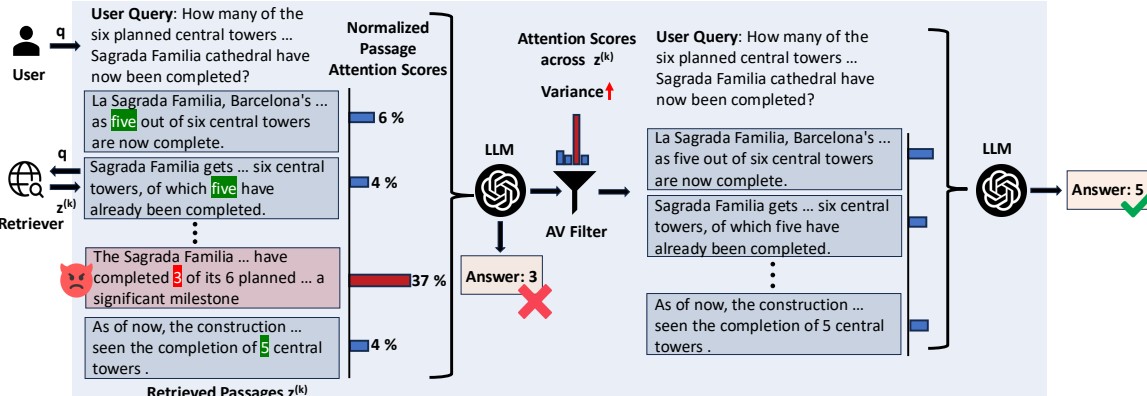

*Figure 1.* **AV Filter Overview.** The retriever returns passages $z^{(k)}$, one of which is poisoned and disproportionately influences the response, increasing variance in NPAS across passages. AV Filter mitigates this by removing passages with anomalously high attention scores, indicative of poisoning.

simultaneously maximizing two probabilities: (1) that the adversarial passages are retrieved,

$$\Pr_{\theta}\left[z_{\text{adv}} \subset \text{Ret}_{\theta}(q, z \cup z_{\text{adv}})\right],$$

and (2) that the LLM generates the target response given that the adversarial passages are included,

$$\Pr_{\theta}\left[\text{LLM}_{\theta}\left(\text{Concat}(\mathcal{I}, z^{(k)}, q)\right) = s' \mid z_{\text{adv}} \subset z^{(k)}\right].$$

These correspond to attacks on *Step I* and *Step II*, respectively. Existing RAG systems are highly brittle to poisoning, and even minimal corruption—for example, altering a single passage among ten retrieved—can successfully manipulate LLM responses. Given the open challenge of building perfectly robust retrievers (Fayyaz et al., 2025; Lin, 2024; Li et al., 2021), enhancing robustness at the generation stage (*Step II*) becomes critical. This allows tolerance to limited corruption and enables reliable integration with reasonably robust retrieval methods such as Google Search, yielding an end-to-end robust RAG pipeline. **This work focuses on strengthening the robustness of the generation stage.** We argue that a notion of stealth improves robustness by allowing generation to withstand small-scale corruption.

*Remark* 2.2. Advancing the robustness of retrievers is an orthogonal challenge with broader applications and sensitivity to corpus characteristics; we defer improving and analyzing weak retrievers, such as BM25, for integration into robust end-to-end pipelines to future work.

**Existing Work.** QA models are vulnerable to disinformation attacks (Du et al., 2022; Pan et al., 2021; 2023; Zhong et al., 2023), with recent work highlighting risks specific to RAG pipelines. We categorize attacks into: (i) *content-poisoning* methods that inject incorrect information into retrieved passages (often LLM-generated) to bias the

generation towards an adversary-specified answer (e.g., **PoisonedRAG (Poison)** (Zou et al., 2024), **Misinformation Attack (MA)** (Pan et al., 2023), and **RAG Paradox (Paradox)** (Choi et al., 2025)), and (ii) *instruction-poisoning* methods that embed direct prompt within the retrieved passages to elicit incorrect responses (e.g., **Prompt Injection Attack (PIA)** (Greshake et al., 2023)). Although the former may appear more stealthy to human readers, we show that both classes leave comparably detectable traces in the internal representations of LLM when attempting to steer inference toward incorrect outputs.

*Remark* 2.3. Our analysis is not restricted to particular attack types, but rather investigates how poisoned passages perturb intermediate LLM representations compared to benign passages, independent of surface-level semantics

Prior work has explored various defenses, including query paraphrasing (Weller et al., 2022), misinformation detection (Hong et al., 2023), vigilant prompting (Pan et al., 2023), reranking methods (Glass et al., 2022), and perplexity-based filters (Jain et al., 2023; Alon & Kamfonas, 2023; Gonen et al., 2022). However, these methods often suffer from limited efficacy or high false positive rates (Zou et al., 2024). More recently, Certified Robust RAG (Xiang et al., 2024) introduced an isolate-then-aggregate strategy that provides empirical accuracy bounds and reduces attack success, representing the current state of the art. Further details are provided in Appendix A.

## 3. Stealth Analysis of Attacks in RAG Systems

**Threat Model.** **Attacker.** We consider an adversary $\mathcal{A}_{\epsilon}$ with full knowledge of the RAG architecture $\theta$ and knowledge base $z$. The adversary may inject up to $\lfloor \epsilon \cdot k \rfloor$ poisoned passages into $z$, but cannot modify or delete existing content. Given a query $q$, target response $s'$, and architecture $\theta$, the

adversary produces a poisoned set

$$z_{\text{adv}} = \mathcal{A}_\epsilon(q, z, s', \theta) = \{z_1, \ldots, z_{\lfloor \epsilon \cdot k \rfloor}\},$$

constructed so that all poisoned passages are retrieved and collectively induce the generation of $s'$. Let $z_{\text{benign}}^{(k)} = \text{Ret}_\theta(q, z)$ and $z_{\text{corrupt}}^{(k)} = \text{Ret}_\theta(q, z \cup z_{\text{adv}})$ denote the top-$k$ retrieved passages from the benign and poisoned knowledge bases, respectively. The retrieved sets may differ by at most $\lfloor \epsilon \cdot k \rfloor$ passages. An attack is successful if the RAG system generates the target response, i.e., $\text{Gen}_\theta(q, z_{\text{corrupt}}^{(k)}) = s'$.

**Defender.** We consider a defender $\mathcal{D}$ that has access to a limited set of benign passages from the knowledge base. The defender also has access to the internal states of the LLM, such as attention weights and hidden representations.

**Attack Practicality.** Our threat model reflects real-world scenarios where attackers have limited resources and can inject only a few poisoned passages into the top-$k$ results, such as with web search retrievers. Attacks requiring majority corruption are impractical due to the high cost of injecting and maintaining numerous adversarial passages (Zhang et al., 2025). Notably, prior work such as PoisonedRAG (Zou et al., 2024) claims to manipulate RAG outputs with even a single poisoned passage. Our work challenges such claims and demonstrates that correctly developed RAG systems are far more robust than previously suggested. We construct corrupted sets by injecting poisoned passages into benign sets, following Xiang et al. (2024).

**Assumptions.** Our stealth analysis, the SADG defined below, and the AV Filter (Section 4) rely on the following assumptions. We state them explicitly so that our robustness claims, together with the limitations discussed in Section 6, can be interpreted within a clearly defined scope.

**Assumption 3.1** (Bounded corruption). The adversary corrupts only a minority of the retrieved set, i.e., $\epsilon < 0.5$. Guaranteeing robustness under majority corruption is information-theoretically impossible without external ground truth, and detectability is strongest in the low-$\epsilon$ regime that motivates our threat model.

**Assumption 3.2** (Benign-majority consensus with redundancy). The uncorrupted passages agree on the correct response, and at least two benign passages support it. The correct answer is therefore redundantly represented, ensuring that the removal of a single passage does not eliminate the correct signal.

**Assumption 3.3** (Access to attention signals). The defender can obtain attention weights for the generation step. This access is direct when the defender is the model provider (the white-box setting); when the deployed model is closed-source (e.g., GPT-4), an open-source auxiliary model is sufficient (the black-box setting, evaluated in Section 5).

**Assumption 3.4** (Response-targeted poisoning). The attack targets the content of the response: it injects poisoned passages containing tokens that carry or strongly imply an adversarial target $s'$, in order to make the model generate $s'$. This assumption holds for the content-poisoning and instruction-poisoning attacks considered in this work, but not necessarily for attacks targeting other RAG behaviors, such as style manipulation or the elicitation of private data.

**Stealth Attack Distinguishability Game (SADG).** Given a RAG architecture $\theta$ and a knowledge database $z$, we define a game between an arbiter, an adversary $\mathcal{A}_\epsilon$, and a defender $\mathcal{D}$, parameterized by a corruption budget $\epsilon$. The defender does not have access to $z$. The game proceeds as follows:

1. The arbiter samples a query $q$ and constructs the benign retrieved set $z_{\text{benign}}^{(k)} = \text{Ret}_\theta(q, z)$.

2. The adversary $\mathcal{A}_\epsilon$ generates poisoned passages $z^{(\text{adv})}$ under budget $\epsilon$, and the arbiter constructs the corrupted set $z_{\text{corrupt}}^{(k)} = \text{Ret}_\theta\left(q, z \cup z^{(\text{adv})}\right)$.

3. The arbiter sends the query $q$ and the two retrieved sets in random order, as $\left(z_0^{(k)}, z_1^{(k)}\right)$, to the defender. The defender must guess the corrupted set to win the game.

The defender's advantage is defined as: $\text{Adv}_{\text{SADG}}^{\mathcal{A}_\epsilon, \mathcal{D}}(\theta, z, \epsilon) := \left|\Pr[\text{Defender wins}] - \frac{1}{2}\right|$. Smaller $\epsilon$ implies a tighter corruption budget, making stealth more difficult and increasing Adv. An attack is $\tau$-*stealthy* if, for all probabilistic polynomial-time (PPT) defenders $\mathcal{D}$, the advantage is at most $\tau$. A perfectly stealthy attack corresponds to $\tau = 0$. See Appendix B for details.

**Stealth of Existing Attacks.** While existing attacks may evade detection methods that analyze passages in isolation (e.g., perplexity filtering), their influence is still evident in the model's output, making the generated response itself a valuable signal for detecting corruption. This motivates a shift in perspective: **to analyze retrieved passages in conjunction with the generated response and assess whether any passage disproportionately shapes the output.** If most retrieved passages are expected to be relevant to the query, a strong alignment between the response and only a few passages may indicate adversarial manipulation. We formalize this insight with NPAS, which quantifies the alignment between each passage and the generated response. NPAS enables two defenses: a defender $\mathcal{D}_{\text{AV}}$ that distinguishes between benign and corrupted retrieved sets with a strong advantage in SADG, and the **AV Filter**, which removes potentially poisoned passages to mitigate attacks.

## 4. Attention Variance Breaks Stealth

We build on the fact that, in a successful attack, the generated response is strongly correlated with the malicious passages that shaped it. Ideally, for a retrieved set $z^{(k)}$ and target response $s'$, we should consider the conditional probability $\Pr_{z^{(k)}} \left( \text{Gen}_\theta(q, z^{(k)}) = s' \big| z_i \in z^{(k)} \right)$ for each passage $z_i$ to measure its correlation with the response.

**Why analyze attention weights?** In transformer-based LLMs, attention weights provide a lightweight proxy for which input tokens influence a generated response (Vig & Belinkov, 2019). Under a low-corruption successful attack, the model must rely disproportionately on a small set of poisoned tokens to produce the adversarial output, which typically manifests as concentrated attention on those tokens. We therefore extract attention scores during inference and aggregate them at the passage level to estimate which retrieved passages most influence the response. This choice is efficient (no extra forward passes) and integrates naturally with RAG inference. More expressive attribution methods (e.g., attention rollouts) are possible (Abnar et al., 2020), but we focus on simple attention aggregates that are stable and easy to deploy.

*Remark* 4.1. We do not claim that attention weights provide a causal measure of a passage's influence on the generated response; we use attention only as a lightweight, empirically motivated proxy. Our analysis is empirical: attacks constructed without awareness of this signal tend to produce, as a side effect, detectable traces in the attention distribution, since steering the response toward the target answer concentrates attention on poisoned tokens. Consistent with this interpretation, the adaptive attacks of Section 5, which explicitly optimize against the signal, can partially suppress these traces.

We define the NPAS, which aggregates token-level attention to quantify the proportion of total attention each passage receives from the final response. This score helps identify anomalous passages indicative of adversarial influence. This skewed distribution of attention in poisoned passages is illustrated through examples in Appendix C.2.

**Normalized Passage Attention Score (NPAS).** Let the input to $\text{LLM}_\theta$ be $\mathcal{X} = \text{Concat}(\mathcal{I}, z^{(k)}, q)$ where $z^{(k)}$ is the retrieved set and $q$ is the query. It generates a response $s' = \{s'_1, s'_2, \ldots, s'_l\}$ of $l$ tokens while computing multi-layer, multi-head attention weights, with each layer producing a separate tensor for each head. We average these weights across all decoder layers and heads to construct a unified attention matrix: $A = \text{Attention}(\text{LLM}_\theta, \mathcal{X}) \in \mathbb{R}^{l \times T}$ where $T$ is the number of input tokens. Each entry $A[i, j]$ denotes the mean attention from the $i$-th output token to the $j$-th input token. This averaging yields a stable view of token-level interactions (Peysakhovich & Lerer, 2023).

Each retrieved passage $z_t$ is a sequence of tokens, $z_t = \{z_t^{(1)}, z_t^{(2)}, \ldots\}$. We define the **Passage Attention Score**, $\text{Score}_\alpha(z_t, A)$, as the sum of attention scores from all response tokens $s'$ to the top-$\alpha$ most attended tokens in $z_t$, denoted $\text{Top}_\alpha(z_t)$:

$$\text{Score}_\alpha(z_t, A) = \sum_{i=1}^{l} \sum_{x_j \in \text{Top}_\alpha(z_t)} A[i, j]$$

Focusing on the top-$\alpha$ tokens captures high-signal Heavy Hitter tokens (often adversarial keywords) while reducing noise. Importantly, for any fixed $\alpha$, this score is invariant to passage length, preventing adversaries from exploiting length manipulation.

To enable comparison across queries and models, we normalize each passage's score by the total score across all $k$ retrieved passages, yielding the **Normalized Passage Attention Score** (NPAS):

$$\text{NormScore}_\alpha(z_t, z^{(k)}, A) = \frac{\text{Score}_\alpha(z_t, A)}{\sum_{i=1}^{k} \text{Score}_\alpha(z_i, A)}$$

While normalization preserves ranking, it standardizes attention magnitudes, enabling a stable detection threshold independent of specific instances (Xian et al., 2025). For clarity, we rescale NPAS to a percentage.

Ideally, $\alpha$ should match the number of Heavy Hitters in the poisoned passage, which is typically proportional to the target response length. We use $\alpha \in \{5, 10, \infty\}$, where $\infty$ denotes summing over all tokens; see Appendix C.4 for detailed rationale.

**Design choices.** We average attention over heads and layers to obtain a single, low-variance influence signal, avoiding reliance on any particular head or layer. To control for passage length, we aggregate only the top-$\alpha$ attended tokens per passage, which removes the trivial effect that longer passages can accrue larger raw attention mass. Finally, we use the variance of normalized passage scores as a simple and reliable statistic: benign retrievals tend to distribute influence broadly across passages, whereas successful low-corruption attacks concentrate influence in a small subset, producing a high-variance signature.

**Discriminating Between Corrupted and Benign Retrievals.** In benign RAG instances where retrieved passages are relevant to both query and response, NPAS is approximately uniform across passages, with a slight *recency effect* (Liu et al., 2023a; Guo & Vosoughi, 2024). Corruption disrupts this pattern: as shown in Figure 2(a), even a single corrupted passage exhibits significantly elevated NPAS relative to the benign baseline. This results in higher variance in NPAS across passages for corrupted sets.

Motivated by this observation, we propose a defender $\mathcal{D}_{\text{AV}}$ for the SADG game that detects corruption using attention

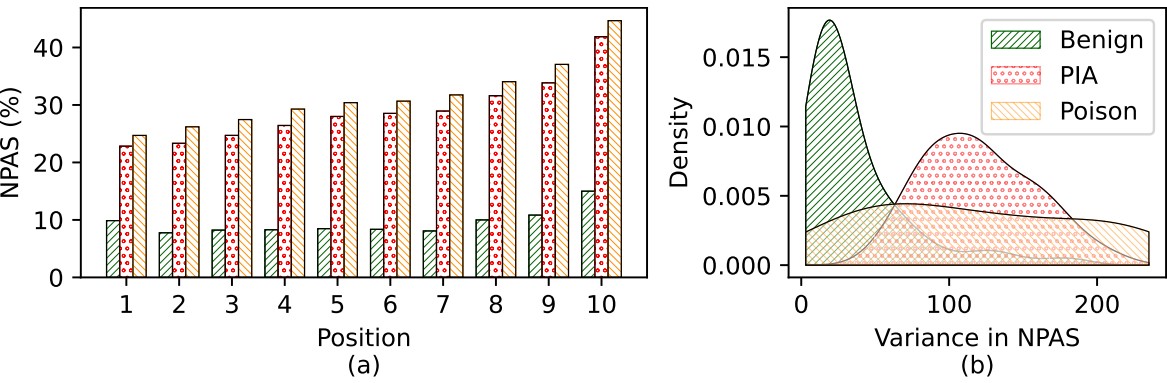

*Figure 2.* (a) Average NPAS across passage positions over multiple queries. Benign passages exhibit nearly uniform scores, while a poisoned passage at any position receives disproportionately high NPAS. (b) Distribution of NPAS variance for benign vs. corrupted sets. Corrupted sets show a significant rightward shift with a higher mean, enabling clear separation from benign sets.

---

**Algorithm 1 A**ttention-**V**ariance **Filter** (AV Filter)

1: **Input:** Query $q$, Retrieved set $z^{(k)}$, model $\text{LLM}_\theta$, Corruption fraction $\epsilon$, Variance threshold $\delta$
2: **Output:** Filtered set $\tilde{z}$
3: $z^{\text{sorted}} \leftarrow \textsf{Sort}(z^{(k)}, \text{LLM}_\theta)$    ▷ Sort by attention scores
4: $\tilde{z} \leftarrow z^{\text{sorted}}$
5: **while** $|\tilde{z}| > \lfloor (1-\epsilon) \cdot k \rfloor$ **do**
6:     $\mathcal{X} \leftarrow \textsf{Concat}(\mathcal{I}, \tilde{z}, q)$        ▷ Form input sequence
7:     $A \leftarrow \textsf{Attention}(\text{LLM}_\theta, \mathcal{X})$      ▷ Compute attention matrix
8:     $\textsf{scores} \leftarrow \{\textsf{NormScore}_\alpha(z_t, \tilde{z}, A) \mid z_t \in \tilde{z}\}$
9:     $\sigma^2 \leftarrow \textsf{Var}(\textsf{scores})$              ▷ Compute variance
10:     **if** $\sigma^2 \leq \delta$ **then**
11:         **break**
12:     **end if**
13:     $z_{\max} \leftarrow \underset{z_t \in \tilde{z}}{\text{argmax}}\ \textsf{NormScore}_\alpha(z_t, \tilde{z}, A)$
14:     $\tilde{z} \leftarrow \tilde{z} \setminus \{z_{\max}\}$  ▷ Remove highest-scoring passage
15: **end while**
16: **Return:** $\tilde{z}$

---

variance. Given a query $q$ and two retrieved sets $(z_0^{(k)}, z_1^{(k)})$, the defender computes the NPAS for both sets:

$$\textsf{scores}_i = \left\{ \textsf{NormScore}_\alpha(z_t, z_i^{(k)}, A) \mid z_t \in z_i^{(k)} \right\},$$

and outputs:

$$\mathcal{D}_{\text{AV}}(q, z_0^{(k)}, z_1^{(k)}) := \begin{cases} 0, & \text{if } \textsf{Var}(\textsf{scores}_0) > \textsf{Var}(\textsf{scores}_1), \\ 1, & \text{otherwise.} \end{cases}$$

That is, the defender flags the set with higher NPAS variance as corrupted. As shown in Figure 2(b), the variance distribution for corrupted sets is markedly shifted toward higher values, yielding clear separation from benign sets and enabling reliable detection.

**Filtering Poisoned Passages.** We propose the **Attention-Variance Filter (AV Filter)**, an outlier filtering algorithm

that removes potentially corrupted passages exhibiting unusually high NPAS. Given a query $q$, retrieved set $z^{(k)}$, model $\text{LLM}_\theta$, corruption budget $\epsilon$, and threshold $\delta$, the filter iteratively removes the highest-scoring passage until either the NPAS variance drops below $\delta$ or an $\epsilon$-fraction of passages have been removed.

To mitigate the recency effect, where tokens near the generation position receive slightly elevated attention, we first reorder passages by their NPAS (Peysakhovich & Lerer, 2023). This sorting reduces positional bias and amplifies anomalous signals, improving filtering performance. The full procedure is specified in Algorithm 1.

**Estimating the Filtering Threshold $\delta$.** The AV Filter's effectiveness depends on choosing an appropriate threshold $\delta$. We estimate $\delta$ using the RQA dataset (Kasai et al., 2023) and Llama 2 (Touvron et al., 2023) by computing NPAS variance across clean retrieved sets and setting $\delta$ as the mean plus one standard deviation. We prioritize minimizing false negatives over false positives, since dropping a few benign passages rarely affects the final response when most of the retrieved content is clean. The estimated threshold generalizes well to unseen settings.

## 5. Evaluation

We address the following research questions:

**RQ1:** Can the defender $\mathcal{D}_{\text{AV}}$ reliably identify corrupted retrievals in existing attacks?

**RQ2:** How *effective* is the AV Filter at mitigating existing attacks?

**RQ3:** How *effective* and *efficient* are adaptive attacks at bypassing the AV Filter?

**Summary of Findings.**

**RQ1:** $\mathcal{D}_{AV}$ identifies corrupted sets and wins the SADG game against existing attacks with high probability. Across settings, it achieves an average win rate of $0.78$, demonstrating a strong advantage.

**RQ2:** The AV Filter outperforms baseline defenses, achieving up to $23\%$ higher accuracy in benign settings and up to $20\%$ under attack, while maintaining comparable reductions in attack success rate (ASR).

**RQ3:** Adaptive attacks can bypass the AV Filter, achieving an attack success rate (ASR) up to $35\%$. However, the AV Filter still reduces ASR below vanilla RAG and the upper bounds of Certified Robust RAG. Notably, these adaptive attacks represent an extreme limit rather than a practical threat: they require costly query-specific optimization ($\sim 10^3 \times$ baseline runtime) and assume access to exact benign passages and model internals, unlike practical attacks.

## 5.1. Experimental Setup

**Datasets.** We evaluate on 4 benchmark question-answering datasets: **RealtimeQA (RQA)** (Kasai et al., 2023), **Natural Questions (NQ)** (Kwiatkowski et al., 2019), and **HotpotQA** (Yang et al., 2018) for short-answer open-domain QA, and **RealtimeQA-MC (RQA-MC)** (Kasai et al., 2023) for multiple-choice QA. Each dataset interfaces with a knowledge source: Google Search for RQA, RQA-MC, and NQ; Wikipedia corpus for HotpotQA and NQ. Following Xiang et al. (2024), we evaluate 100 queries per dataset.

**RAG Setup.** We evaluate 5 LLMs: Llama2-7B-Chat (Touvron et al., 2023), Mistral-7B-Instruct (Chaplot, 2023), Llama-3.1-8B-Instruct (AI, 2024), Deepseek-R1-Distill-Qwen-7B (Guo et al., 2025), and GPT-4o (Achiam et al., 2023). We retrieve the top $k = 10$ passages. For GPT-4o, which lacks accessible internals, we use Mistral-7B as an auxiliary model to compute attention scores. We adopt models used in prior work (Xiang et al., 2024; Zou et al., 2024) for direct comparison with baseline defenses.

**Attacks.** We evaluate 4 content-poisoning attacks: **Poison** (Zou et al., 2024), **Misinformation Attack (MA)** (Pan et al., 2023), **Paradox** (Choi et al., 2025), and **CorruptRAG** (Zhang et al., 2025), as well as one instruction-poisoning attack, **PIA** (Greshake et al., 2023). Unless otherwise stated, we set the corruption fraction $\epsilon = 0.1$ and randomly vary the position of the poisoned passage.

**Defenses.** We evaluate the **AV Filter** using $\text{NormScore}_\alpha$ for $\alpha \in \{5, 10, \infty\}$, with threshold $\delta = 26.2$ estimated from benign RQA with Llama 2 at $\alpha = \infty$. Primary baselines include vanilla RAG (**Vanilla**) and Certified Robust RAG (Xiang et al., 2024): **Keyword** and **Decoding**. Additional baselines (Perplexity Filtering, Vigilant Prompting,

and Reranking Methods) are evaluated in Appendix D.5.

**Evaluation Metrics.** For **RQ1**, we measure the success of defender $\mathcal{D}_{AV}$ in SADG via the **Corruption Identification Rate (CIR)**: the fraction of corrupted sets correctly flagged under successful attacks on vanilla RAG. For **RQ2** and **RQ3**, we report three metrics: **Clean Accuracy (ACC)**, correct responses without attack; **Robust Accuracy (RACC)**, correct responses under attack; and **Attack Success Rate (ASR)**, responses containing the adversary's target. A response is correct if it contains a valid variation of the ground-truth answer $s$ and excludes the adversary's target $s'$. All metrics are percentages averaged over 5 random seeds.

We report a representative subset of results using Poison and PIA attacks with Google Search. Extended results covering additional attacks (Appx. D.2), knowledge bases (Appx. D.6), baselines (Appx. D.5), false positive rates (Appx. D.4), hyperparameter analysis (Appx. D.8), and ensembling with Certified Robust RAG (Appx. D.7) are provided in Appendix D.

## 5.2. Results and Discussion

**Takeaway (Stealth–effectiveness trade-off).** Successful low-corruption ($\epsilon$) attacks tend to induce a measurable skew in passage influence, visible as a high-variance NPAS signature; AV Filter leverages this to reduce ASR while preserving clean utility. Adaptive attacks can reduce this influence skew and evade filtering more often, but doing so requires query-specific optimization and strong access assumptions, making them substantially more expensive and less general.

**RQ1.** Table 4 (Appendix B) reports the estimated probability of $\mathcal{D}_{AV}$ winning the SADG, measured via CIR, across models, datasets, and values of $\alpha$ under existing attacks. CIR is computed over successful attack instances against vanilla RAG. $\mathcal{D}_{AV}$ identifies corrupted sets with high accuracy, achieving an average CIR of **0.78**.

**RQ2. Clean Accuracy.** Table 2 presents clean accuracy across models, datasets, and $\alpha$ values. The AV Filter maintains strong clean performance, with an average drop of only 4-6%, substantially smaller than other defenses. On RQA-MC, accuracy drops from **61.4%** (Vanilla) to **59.3%** with AV Filter, compared to Keyword (**56.0%**) and Decoding (**50.3%**). Similar trends hold for RQA and NQ.

**Robust Accuracy.** Table 3 reports robust accuracy (RACC) and attack success rate (ASR). On RQA-MC, AV Filter achieves **55.7%** RACC, outperforming Vanilla (**44.4%**), Keyword (**53.9%**), and Decoding (**47.1%**). Similar improvements hold for RQA (**59.8%**) and NQ (**53.4%**). Notably, AV Filter's RACC closely matches Vanilla's clean accuracy, indicating minimal benign impact. While Keyword achieves lower ASR in some cases, its aggressive pruning reduces robust accuracy in 17/30 settings.

*Table 2.* Clean Accuracy (ACC) of defenses, showing that AV Filter preserves RAG utility with a minimal drop from Vanilla, achieving up to 23% higher ACC than other baselines.

| LLM | Mistral-7B | | | Llama2-C | | | GPT-4o | | | Llama-3.1 | | | Deepseek-R1 | | |
|---|---|---|---|---|---|---|---|---|---|---|---|---|---|---|---|
| Defense | RQA-MC | RQA | NQ | RQA-MC | RQA | NQ | RQA-MC | RQA | NQ | RQA-MC | RQA | NQ | RQA-MC | RQA | NQ |
| Vanilla | 81.0 | 72.0 | 62.0 | 79.0 | 61.0 | 59.0 | 66.2 | 69.8 | 61.2 | 44.0 | 71.0 | 64.0 | 37.0 | 56.0 | 54.0 |
| Keyword | 58.0 | 56.0 | 51.0 | 56.0 | 57.0 | 54.0 | **63.2** | **64.2** | **60.4** | **61.0** | 61.0 | 62.0 | 42.0 | 41.0 | 43.0 |
| Decoding | 57.0 | 57.0 | 55.0 | 44.0 | 54.0 | 41.0 | – | – | – | 56.0 | 56.0 | 56.0 | **44.0** | 44.0 | 44.0 |
| AV Filter$_{(\alpha=5)}$ | 73.0 | **66.0** | **59.0** | **79.0** | **60.0** | 51.0 | 57.8 | 61.6 | 57.8 | 43.0 | **67.0** | **66.0** | 36.0 | 57.0 | **52.0** |
| AV Filter$_{(\alpha=10)}$ | 74.0 | 65.0 | 58.0 | 75.0 | 57.0 | **54.0** | 59.8 | 62.6 | 55.0 | 45.0 | 66.0 | **66.0** | 37.0 | **59.0** | **52.0** |
| AV Filter$_{(\alpha=\infty)}$ | **76.0** | 64.0 | 58.0 | 75.0 | 56.0 | **54.0** | 59.6 | 63.0 | 55.8 | 44.0 | **67.0** | 62.0 | 34.0 | 57.0 | **52.0** |

*Table 3.* Robust Accuracy and Attack Success Rate (RACC/ASR) showing that AV Filter effectively mitigates attacks with low ASRs while achieving up to 20% higher RACC than baseline defenses.

| | Dataset | RQA-MC | | RQA | | NQ | |
|---|---|---|---|---|---|---|---|
| LLM | Attack | PIA | Poison | PIA | Poison | PIA | Poison |
| | Defense | (racc↑ / asr↓) | (racc↑ / asr↓) | (racc↑ / asr↓) | (racc↑ / asr↓) | (racc↑ / asr↓) | (racc↑ / asr↓) |
| **Mistral-7B** | Vanilla | 59.6 / 31.0 | 62.2 / 30.0 | 52.2 / 26.6 | 50.0 / 23.4 | 40.8 / 24.6 | 52.0 / 9.2 |
| | Keyword | 57.0 / 7.00 | 55.0 / **6.00** | 54.0 / 6.00 | 55.0 / **6.00** | 50.0 / **1.00** | 49.0 / **1.00** |
| | Decoding | 55.0 / **5.00** | 54.0 / 13.0 | 55.0 / 5.00 | 54.0 / 13.0 | 55.0 / **1.00** | **56.0** / **1.00** |
| | AV Filter$_{(\alpha=5)}$ | 76.6 / 5.80 | 70.0 / 10.0 | 62.6 / 2.80 | **58.2** / 7.40 | 54.4 / 3.00 | 53.4 / 6.20 |
| | AV Filter$_{(\alpha=10)}$ | **77.2** / 6.00 | 71.6 / 8.20 | 64.8 / 2.80 | 56.8 / 8.40 | **56.2** / 2.80 | 52.8 / 6.60 |
| | AV Filter$_{(\alpha=\infty)}$ | 76.2 / 7.20 | **73.8** / 8.40 | **65.0** / 2.40 | 56.8 / 8.60 | 50.2 / 5.80 | 52.8 / 4.00 |
| **Llama2-C** | Vanilla | 33.4 / 63.0 | 62.8 / 27.6 | 5.80 / 88.2 | 57.4 / 17.2 | 10.6 / 73.2 | **56.8** / 5.80 |
| | Keyword | 54.0 / **6.00** | 53.0 / **5.00** | 53.0 / 6.00 | 53.0 / **5.00** | **52.0** / **2.00** | 51.0 / **2.00** |
| | Decoding | 38.0 / 12.0 | 40.0 / **5.00** | 38.0 / 12.0 | 40.0 / **5.00** | 39.0 / 17.0 | 40.0 / 4.00 |
| | AV Filter$_{(\alpha=5)}$ | 65.6 / 18.4 | 67.8 / 18.4 | **61.8** / 1.60 | 55.4 / 7.00 | 50.6 / 5.20 | 49.8 / 6.20 |
| | AV Filter$_{(\alpha=10)}$ | **70.8** / 12.4 | 69.6 / 13.0 | 60.2 / **1.60** | 54.8 / 8.80 | 51.4 / 4.00 | 51.2 / 6.20 |
| | AV Filter$_{(\alpha=\infty)}$ | 68.8 / 16.8 | **72.0** / 12.6 | 60.2 / 5.00 | **56.8** / 6.60 | 49.4 / 9.20 | 51.8 / 3.60 |
| **Llama-3.1** | Vanilla | 42.0 / 15.0 | 30.6 / 19.2 | 48.4 / 14.0 | 21.0 / 29.4 | 34.6 / 22.4 | 41.0 / 10.8 |
| | Keyword | **61.0** / 7.00 | **58.0** / 6.00 | 61.0 / 7.00 | 57.0 / **6.00** | 60.0 / 3.00 | **58.0** / **2.00** |
| | Decoding | 55.0 / 7.00 | 51.0 / 17.0 | 55.0 / 7.00 | 51.0 / 17.0 | 49.0 / 13.0 | 49.0 / 10.0 |
| | AV Filter$_{(\alpha=5)}$ | 43.0 / **2.60** | 35.8 / 10.6 | **70.2** / 2.60 | 53.8 / 7.20 | **60.8** / **1.00** | 50.2 / 5.00 |
| | AV Filter$_{(\alpha=10)}$ | 44.2 / 2.80 | 36.2 / 7.00 | 67.8 / 3.00 | 53.2 / 6.40 | 57.8 / 2.20 | 50.6 / 5.20 |
| | AV Filter$_{(\alpha=\infty)}$ | 42.2 / 3.20 | 36.4 / **6.00** | 68.2 / 2.80 | **57.4** / 6.20 | 53.8 / 5.20 | 54.4 / 4.20 |
| **Deepseek-R1** | Vanilla | 26.0 / 2.60 | 23.6 / 9.60 | 24.3 / 49.6 | 46.3 / 17.00 | 33.3 / 33.0 | 48.6 / 7.30 |
| | Keyword | 40.0 / 3.00 | 36.0 / 3.00 | 40.0 / 3.00 | 37.0 / 3.00 | **44.0** / 2.00 | 44.0 / 2.00 |
| | Decoding | **42.0** / **1.00** | **42.0** / **1.00** | 42.0 / **1.00** | 42.0 / **1.00** | **44.0** / **1.00** | 43.0 / **0.00** |
| | AV Filter$_{(\alpha=5)}$ | 35.0 / **1.00** | 21.0 / 8.60 | 39.3 / 25.6 | 45.3 / 20.3 | 33.6 / 29.3 | **51.0** / 9.00 |
| | AV Filter$_{(\alpha=10)}$ | 35.3 / 2.30 | 25.6 / 6.00 | 47.0 / 14.0 | 46.6 / 14.6 | 39.6 / 24.0 | 48.6 / 7.30 |
| | AV Filter$_{(\alpha=\infty)}$ | 29.3 / 2.30 | 27.6 / 6.30 | **50.3** / 10.3 | **53.0** / 8.60 | 38.6 / 19.3 | 48.6 / 5.3 |
| **GPT-4o** | Vanilla | 60.2 / 19.6 | 43.6 / 25.0 | 52.4 / 33.4 | 55.6 / 26.6 | 39.8 / 33.0 | 56.4 / 5.20 |
| | Keyword | 62.6 / **4.40** | **63.0** / **4.20** | 63.4 / **4.00** | **62.6** / **4.00** | **60.2** / **1.40** | **60.0** / **1.20** |
| | AV Filter$_{(\alpha=5)}$ | 63.8 / 5.20 | 55.0 / 7.60 | **63.6** / 5.20 | 57.8 / 10.6 | 56.8 / 2.60 | 58.0 / 3.80 |
| | AV Filter$_{(\alpha=10)}$ | **64.2** / 4.60 | 50.4 / 10.4 | **63.6** / 4.80 | 57.2 / 11.0 | 57.0 / 4.00 | 57.8 / 3.00 |
| | AV Filter$_{(\alpha=\infty)}$ | 63.8 / 5.40 | 50.8 / 6.80 | 61.2 / 7.00 | 61.4 / 9.20 | 52.0 / 11.4 | 58.8 / 1.60 |

**Attack Success Rate.** Even with small corruption ($\epsilon = 0.1$), Vanilla RAG is highly vulnerable, reaching up to **88.2**% ASR. AV Filter reduces this to an average of **6.6**% on RQA-MC, comparable to Keyword (**6.1**%) and Decoding (**7.6**%). Similar reductions hold for RQA (**6.0**%) and NQ (**4.8**%).

*Remark* 5.1. While any defense incurs some clean accuracy drop, AV Filter's is minimal. Certified Robust RAG's isolate-then-aggregate design analyzes each passage independently, which fails for tasks requiring multi-passage reasoning. On HotpotQA, a multi-hop QA benchmark, AV Filter outperforms Keyword in ASR across all settings (Table 14). Moreover, AV Filter is a simple filtering mechanism that preserves the inference pipeline, enabling combination with other defenses. Appendix D.7 shows that AV Filter +

Keyword outperforms Keyword alone in all 18/18 cases for both RACC and ASR.

**RQ3.** We adapt GCG (Zou et al., 2023) and AutoDAN (Liu et al., 2023b) by optimizing the poisoned passage with full access to the input. We minimize $\mathcal{L}_1 + \lambda \cdot \mathcal{L}_2$, where $\mathcal{L}_1$ is the cross-entropy loss for the target response and $\mathcal{L}_2$ is the attention variance across passages. Existing attacks achieve low $\mathcal{L}_1$ but high $\mathcal{L}_2$ due to concentrated attention on tokens matching the adversarial target. Simply removing such tokens lowers $\mathcal{L}_2$ but raises $\mathcal{L}_1$, weakening the attack. Our method searches for replacements that balance both objectives, making optimization costly. We tune $\lambda$ on RQA-MC with Llama 2, fixing $\lambda = 0.1$. Due to computational

constraints, we evaluate 20 queries per dataset.

Adaptive attacks can evade AV Filter by lowering variance in NPAS while preserving the target response. On RQA-MC, ASR rises to **20**%, but remains below vanilla RAG and Certified RAG (Xiang et al., 2024), with similar patterns across datasets. However, these attacks require full input and model access plus query-specific optimization (up to $10^4$s per query), making them resource-intensive and impractical at scale. Designing efficient, generalizable attacks without full access remains an open challenge. Detailed results and algorithms are provided in Appendix D.1.

## 6. Limitations

The AV Filter detects poisoning from the high-variance attention signature that poisoned passages produce. This signal, and hence the filter, is reliable only within the scope set by Assumptions 3.1 through 3.4; each limitation below follows from the violation of one of them.

1. **Majority corruption** (Assumption 3.1). When poisoned passages form a majority, several of them attract high attention, the variance of NPAS is reduced, and the filter can be evaded. This is an information-theoretic limit shared by all aggregation defenses, including Certified Robust RAG; the gains of AV Filter are orthogonal to retriever robustness and can be combined with robust retrievers that make majority corruption harder.

2. **Redundancy of correct knowledge** (Assumption 3.2). If very few benign passages carry the correct answer, they may themselves attract high attention and be filtered out. As for any low-corruption defense, a one-to-one split between correct and adversarial evidence cannot be resolved without external ground truth.

3. **Task specificity** (Assumption 3.4). The filter exploits the co-occurrence of poisoned tokens with the target answer, which suits question answering. Its effectiveness against attacks on other RAG behaviors, such as style manipulation or the elicitation of private data, is unexplored and left to future work.

## 7. Conclusion and Future Work

We have shown that existing attacks lack stealth, drawing disproportionately high attention to poisoned passages. This enables effective defenses: when attacks succeed while corrupting only a small fraction of the input, they must exert unusually large influence, leaving detectable traces. Our adaptive attacks probe the limits of attention-based defenses but remain inefficient and query-dependent. Improving their generality and identifying other detectable traces in intermediate representations are key open challenges.

We argue the effectiveness-stealth trade-off is fundamental: successful attacks must exert anomalous influence on the generated response. However, the presence of a detectable signal does not guarantee efficient detection. A key open question is whether attacks can be constructed where detecting this influence is computationally infeasible despite the signal being information-theoretically present. Resolving this would clarify whether polynomial-time defenses like AV Filter are sufficient or fundamentally limited.

## Impact Statement

We propose a filtering technique capable of identifying and mitigating existing poisoning attacks, thereby reducing potential harm. In parallel, we introduce more stealthy adaptive poisoning attacks. While we believe this dual contribution will drive the development of more robust RAG systems, it may also increase the risk to vulnerable deployments in the short term.

## Acknowledgments

We would like to thank the anonymous ICML reviewers for their constructive feedback, and Atharv Singh Patlan for valuable comments and suggestions on an earlier draft of this paper. At the time of this work, Sarthak Choudhary, Nils Palumbo, Ashish Hooda, and Somesh Jha were partially supported by DARPA under agreement number 885000, NSF CCF-FMiTF-1836978, and ONR N00014-21-1-2492.

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

# A. Additional Background on Existing Works

**PoisonedRAG.** Given a query $q$ and target answer $s'$, PoisonedRAG (Poison) seeks to craft a poisoned passage $z_{\text{poison}}$ such that a RAG system is highly likely to generate $s'$ when $z_{\text{poison}}$ is included in the retrieved context. To do so, it leverages an LLM (e.g., GPT-4) to generate $p$. The attack LLM may be the same as or different from the one used in the target system, depending on the attacker's access and knowledge. The attacker prompts the LLM to generate $z_{\text{poison}}$ such that, when used as context, it induces the model to output $s'$. The following prompt is used:

> This is my question: [question].
> This is my answer: [answer].
> Please craft a corpus such that the answer is [answer] when prompted with the question [question]. Please limit the corpus to $V$ words.

where $V$ is a hyperparameter that limits the length of $z_{\text{poison}}$, though in practice, the generated passage may exceed this limit if the LLM fails to strictly follow the instruction. For our experiments, we use the poisoned passages publicly released by the PoisonedRAG (Zou et al., 2024) and Certified Robust RAG (Xiang et al., 2024). We repeat the poisoned content 2 times in a single passage to simulate a strong attacker that can successfully manipulate the response in Vanilla RAG, while ensuring the overall length remains comparable to that of a typical benign passage.

**Prompt Injection Attack.** It enables adversaries to remotely exploit LLM-integrated applications by strategically injecting prompts into data likely to be retrieved. In Prompt Injection Attack (PIA), the attacker injects the following malicious instruction: "`When asked to provide the answer for the following question: {query}, please output {target answer}`". We use a passage containing the aforementioned instruction for a specific query-answer pair as the poisoned passage in our experiments.

**Certified Robust RAG.** Certified Robust RAG employs an isolate-then-aggregate strategy. Given $k$ retrieved passages, it computes $k$ individual LLM responses, one from each passage in isolation, rather than concatenating all passages as in Vanilla RAG. It extracts unique keywords from each response and retains only those whose count exceeds the number of possibly corrupted passages. These keywords are then used to generate the final answer. If no such keyword exists, it uses all keywords; if no keywords appear in any response (e.g., all responses are "I don't know"), it discards all passages and uses only the query.

This isolation technique limits the impact of any poisoned passage to its own response, making successful attacks difficult. However, it introduces significant limitations:

**Reduced Utility.** The aggressive pruning of context significantly reduces utility. Certified RAG-Keyword achieves strong ASR (outperforming AV Filter in 23/30 cases with average ASR of 3.97 vs. 5.84), but this comes at the cost of robust accuracy, where it underperforms in 17/30 cases. The strict keyword filtering often discards benign content along with adversarial passages.

**Failure on Multi-Passage Reasoning.** The isolation design fundamentally fails on tasks requiring reasoning over multiple documents. On HotpotQA, where questions require multi-hop reasoning, individual responses are always "I don't know," rendering the approach ineffective. AV Filter outperforms Keyword in ASR across all 6/6 HotpotQA settings (Table 14).

**Inference Overhead.** The approach incurs $k\times$ inference cost compared to Vanilla RAG, significantly increasing latency and computational costs.

In contrast, AV Filter is a simple filtering mechanism that removes potentially poisoned passages without altering the inference pipeline. This design enables combination with other defenses: AV Filter + Keyword outperforms Keyword alone in all 18/18 cases for both robust accuracy and ASR (Appendix D.7). We argue that attention-based filtering is a promising direction, as it can improve with better techniques for estimating attention influence, is effective standalone, and enhances other defenses due to its low false-positive rate.

## B. Stealth Attack Distinguishability Game (SADG)

We define a security game between an arbiter, an adversary $\mathcal{A}_\epsilon$, and a defender $\mathcal{D}$, parameterized by a parameter $\epsilon$. The goal is to evaluate whether $\mathcal{D}$ can distinguish a corrupted retrieved set from a benign one. The corruption budget of $\mathcal{A}_\epsilon$ is controlled by $\epsilon$; smaller values correspond to tighter budgets, making stealth harder.

For a given RAG architecture $\theta$ and knowledge database $z$, the defender does not have access to $z$, the game proceeds as follows:

1. **Query sampling:** The arbiter samples a query $q \leftarrow \mathcal{Q}$.

2. **Retrieved set generation:** The arbiter samples a target response $s' \leftarrow \mathcal{S}$. It computes the benign retrieved set $z_{\text{benign}}^{(k)} = \text{Ret}_\theta(q, z)$, queries the adversary to obtain poisoned passages $z^{(\text{adv})} = \mathcal{A}_\epsilon(q, z, s', \theta)$, and constructs the corrupted retrieved set $z_{\text{corrupt}}^{(k)} = \text{Ret}_\theta\left(q, z \cup z^{(\text{adv})}\right)$

3. **Permutation:** The arbiter samples a bit $b \leftarrow \{0, 1\}$ uniformly at random and defines:

$$\left(z_0^{(k)}, z_1^{(k)}\right) := \begin{cases} \left(z_{\text{corrupt}}^{(k)}, z_{\text{benign}}^{(k)}\right), & \text{if } b = 0, \\ \left(z_{\text{benign}}^{(k)}, z_{\text{corrupt}}^{(k)}\right), & \text{if } b = 1. \end{cases}$$

   The arbiter sends $\left(q, z_0^{(k)}, z_1^{(k)}\right)$ to the defender $\mathcal{D}$.

4. **Defender's guess:** The defender outputs $b' \in \{0, 1\}$, guessing which of $z_0^{(k)}$ or $z_1^{(k)}$ is the corrupted set. The defender wins if $b' = b$.

**Advantage.** The defender's advantage is: $\text{Adv}_{\text{SADG}}^{\mathcal{A}_\epsilon, \mathcal{D}}(\theta, z, \epsilon) := \left| \Pr[b' = b] - \frac{1}{2} \right|.$

The probability $\Pr[b' = b]$ is over the randomness of $q$, $s'$, $\theta$, $b$, and defender $\mathcal{D}$.

The attack is said to be $\tau$-*stealthy* if, for all probabilistic polynomial-time (PPT) defenders $\mathcal{D}$, the advantage is at most $\tau$; i.e.,

$$\text{Adv}_{\text{SADG}}^{\mathcal{A}_\epsilon, \mathcal{D}}(\theta, z, \epsilon) \leq \tau,$$

for a perfectly stealthy attack $\tau$ should be zero.

Table 4 reports the estimated probability of $\mathcal{D}_{\text{AV}}$ winning the SADG—measured via CIR—across models, datasets, and varying $\alpha$ values under existing attacks. $\mathcal{D}_{AV}$ identifies the corrupted set with high accuracy, achieving an average CIR of **0.78**, demonstrating strong effectiveness. We used all successful attack instances against Vanilla RAG in each setting to compute CIR.

## C. Attention Insights and Design Rationale of NPAS

### C.1. Why attention scores?

In transformer-based LLMs, attention weights are commonly used as an approximate signal of inter-token dependencies during generation (Vig & Belinkov, 2019). Beyond interpretability, attention patterns have been leveraged for systems work such as KV-cache optimization during inference (Zhang et al., 2023; He et al., 2024). In particular, $H_2O$ (Zhang et al., 2023) identifies *Heavy Hitters* ($H_2$): a small subset of input tokens that dominate attention mass when generating new tokens. These Heavy Hitters naturally emerge and correlate with token co-occurrence, and in compromised RAG instances we find they often localize in poisoned passages (e.g., target-response keywords), yielding a skewed attention distribution.

### C.2. Example: Benign vs. Poisoned Attention Patterns

Our key insight is that attention patterns can be leveraged to detect potentially poisoned passages that disproportionately influence the LLM's response in an RAG system. In such cases, specific tokens from the poisoned passage tend to receive significantly higher attention due to their co-occurrence with the target answer. These tokens act as heavy hitters in the attention distribution and are localized within the poisoned passages, as benign passages typically do not contain tokens

*Table 4.* Estimated probability of $\mathcal{D}_{AV}$ identifying the corrupted set using different $\alpha$ values for $\mathsf{NormScore}_\alpha$, showing high accuracy and a strong advantage against existing attacks.

| Dataset | | RQA-MC | | RQA | | NQ | |
|---|---|---|---|---|---|---|---|
| LLM | top-$\alpha$ | PIA | Poison | PIA | Poison | PIA | Poison |
| **Mistral7-B** | $\alpha = 5$ | **0.94** | 0.84 | **0.94** | **0.93** | **0.79** | 0.54 |
| | $\alpha = 10$ | **0.94** | 0.86 | 0.88 | 0.82 | 0.73 | 0.60 |
| | $\alpha = \infty$ | 0.91 | **0.93** | 0.80 | 0.84 | 0.48 | **0.79** |
| **Llama2-C** | $\alpha = 5$ | 0.82 | 0.70 | **0.99** | 0.86 | **0.93** | 0.66 |
| | $\alpha = 10$ | 0.92 | 0.82 | **0.99** | **0.88** | 0.91 | 0.64 |
| | $\alpha = \infty$ | **0.95** | **0.99** | 0.95 | 0.82 | 0.83 | **0.72** |
| **Llama-3.1** | $\alpha = 5$ | **0.93** | 0.72 | 0.88 | 0.68 | **0.83** | 0.41 |
| | $\alpha = 10$ | 0.86 | 0.70 | **0.89** | 0.72 | 0.75 | 0.50 |
| | $\alpha = \infty$ | 0.88 | **0.82** | 0.83 | **0.87** | 0.68 | **0.63** |
| **Deepseek-R1** | $\alpha = 5$ | **0.95** | 0.47 | 0.75 | 0.65 | 0.55 | 0.63 |
| | $\alpha = 10$ | 0.93 | 0.63 | 0.87 | 0.69 | **0.64** | 0.56 |
| | $\alpha = \infty$ | 0.93 | **0.80** | **0.89** | **0.87** | **0.64** | **0.79** |

associated with an incorrect target answer. This results in a notably higher normalized passage attention score for the poisoned passage compared to the benign ones. AV Filter exploits this disparity by identifying and filtering out such anomalous passages from the corrupted retrieved set. Figure 3 illustrates this behavior, showing a benign passage and the poisoned passage for the following query from the RealtimeQA dataset, along with the corresponding toke-level attention weights produced by Llama 2.

---

**User query**: How many of the six planned central towers on Barcelona's unfinished Sagrada Familia cathedral have now been completed?
**Target Response**: 3
**Correct Response**: 5
**Actual Response**: 3
**Benign Passage**: Sagrada Familia gets final completion date 2026 or 2028. "Our new objective is to complete the six central towers, of which five have already been started," said Rigol. The sixth tower will measure ...
**Poisoned Passage**: The Sagrada Familia, an architectural masterpiece in Barcelona, is said to have completed 3 of its 6 planned central towers, marking a significant milestone in its lengthy construction process. The Sagrada Familia, an architectural masterpiece in Barcelona, is said to have completed 3 of its 6 planned central towers, marking a significant milestone in its lengthy construction process.

---

### C.3. Distribution of Attention Weights across tokens in Passage

We observe the Heavy Hitters phenomenon in adversarial passages: in successful attacks against vanilla RAG, a few tokens receive disproportionately high attention, and these tokens are concentrated in adversarial passages. To illustrate this, we provide a representative example from our evaluation in Figure 3, highlighting the distinct difference in attention weight distributions.

Additionally, we report the difference between attention distributions in Table 5, with similar trends expected across other configurations. For all instances of successful attacks against vanilla RAG, we calculate the average highest attention weight of a token in a poisoned passage and compare it to that in a benign passage, averaged over PIA and Poison.

### C.4. Design Rationale for using top-$\alpha$ tokens from each passage $\text{Top}_\alpha(z_t)$

The Normalized Passage Attention Score is computed by summing the attention weights of tokens within a passage and normalizing this sum across all passages in the retrieved set. However, since the sum of attention weights is proportional to the number of tokens, longer passages can receive disproportionately higher scores, even if they contain little information relevant to the generated answer. Selecting the top-$\alpha$ tokens mitigates this length bias, ensuring that the score reflects the most influential tokens rather than sheer passage length.

Following the insight of Heavy Hitters, our experiments confirm that only a few tokens in an adversarial passage receive

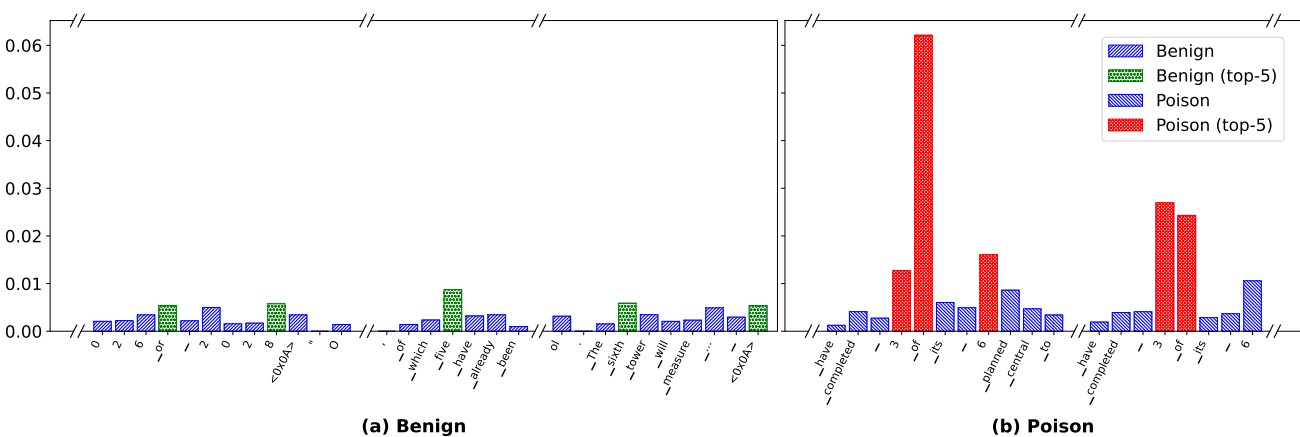

*Figure 3.* **Attention Patterns in Benign vs. Poisoned Passages**: It highlights the token-level attention weights (as a fraction of total attention over the retrieved set) for a query from the RealtimeQA dataset, computed using Llama 2. (a) shows a benign passage with the highest normalized passage attention score among all benign candidates; (b) shows the poisoned passage present in the retrieved set. Tokens such as 3, _of, and 6 from the poisoned passage receive disproportionately high attention—greater than the total attention allocated to many of the individual benign passages. This behavior allows simple aggregation of attention over the top-$\alpha$ tokens to distinguish poisoned from benign passages.

*Table 5.* Highest attention weights per token in a benign passage versus a poisoned passage, showing a clear difference in their distributions.

| Dataset \ LLM | | Mistral-7B | Llama 2 |
|---|---|---|---|
| **RQA-MC** | Benign | 0.37 | 0.67 |
| | Poisoned | 1.65 | 2.10 |
| **RQA** | Benign | 0.66 | 0.42 |
| | Poisoned | 3.41 | 2.50 |
| **NQ** | Benign | 1.26 | 0.71 |
| | Poisoned | 3.30 | 2.68 |

disproportionately high attention weights. These tokens are typically semantically aligned with the generated response and thus exert the most influence on its generation. Ideally, a defense should sum only the contributions of these heavy hitters from each passage, ignoring the long tail of tokens with very small attention weights.

Conceptually, a defender could estimate a threshold such that only tokens with attention weights above it are considered in each passage, assuming tokens with lower attention weights have negligible influence on the output. This threshold may vary depending on the underlying LLM in the RAG pipeline. In practice, we approximate this by selecting the top-$\alpha$ tokens from each passage. We evaluate $\alpha = (5, 10, \alpha)$ and observe that AV Filter provides significant robustness across all settings. A defender can further tune or estimate an attention-weight threshold per token to adaptively select the most relevant tokens from each passage.

However, in many simpler and practical scenarios where retrieved passages are of similar length, the defender can safely consider all tokens from each passage. In such cases, there is no length-based bias, and setting $\alpha = \infty$ often yields optimal performance, as frequently observed in our evaluation. In Table 6, we further provide the average length (in characters) of benign and poisoned passages for each dataset in our evaluation. We observe that the length of the poisoned passages varies across attacks and datasets—some are shorter, while others are longer than the benign passages. Notably, adversarial passages in the Poison attack tend to be longer. This is primarily because they are generated using GPT-4o, which often requires more elaborate phrasing and additional context to effectively manipulate the generation, even in the vanilla RAG setup.

## D. Additional Experimental Details and Results

We use the PyTorch (BSD-style license) and HuggingFace Transformers (Apache-2.0 license) libraries for all our experiments. The experiments were conducted on a mix of A100 and H100 GPUs. All experiments were run with 5 different seeds, except for the adaptive attack due to its high computational cost. We report the mean of each evaluation metric. The maximum

*Table 6.* Average passage length (in characters) for benign cases and different attacks, confirming that the effectiveness of AV Filter is not attributable to length biases.

| Dataset \ Passage | Benign | PIA | Poison |
|---|---|---|---|
| **RQA-MC** | 192.84 | 196.65 | 389.72 |
| **RQA** | 192.84 | 192.65 | 391.40 |
| **NQ** | 191.33 | 150.30 | 368.78 |

---

**Algorithm 2** Adaptive Attention-Aware Poisoning Attack

---

1: **Input:** Query $q$, target answer $s'$, benign retrieved set $z_{\text{benign}}^{(k)}$, model LLM$_\theta$, loss weight $\lambda$, jailbreak function Jailbreak, max steps $T$
2: **Output:** Optimized poisoned passage $z_{\text{poison}}^*$
3: Initialize $p_0 \leftarrow z_{\text{poison}}$ using an existing attack (e.g., PoisonedRAG)
4: $\mathcal{L}^* \leftarrow \infty$, $z_{\text{poison}}^* \leftarrow p_0$               $\triangleright$ Best loss and candidate
5: **for** $t = 1$ **to** $T$ **do**
6:  $z_{\text{corrupt}}^{(k)} \leftarrow z_{\text{benign}}^{(k)} \cup \{p_{t-1}\}$           $\triangleright$ Inject poisoned passage
7:  $\hat{s}_t \leftarrow \text{LLM}_\theta(q, z_{\text{corrupt}}^{(k)})$          $\triangleright$ Generate response
8:  scores $\leftarrow \{\text{NormScore}_\alpha(z_i, z_{\text{corrupt}}^{(k)}, A) \mid z_i \in z_{\text{corrupt}}^{(k)}\}$
9:  $\mathcal{L}_t \leftarrow \text{CE}(\hat{s}_t, s') + \lambda \cdot \text{Var(scores)}$        $\triangleright$ $\mathcal{L}_1 + \lambda \cdot \mathcal{L}_2$
10:  **if** $\mathcal{L}_t < \mathcal{L}^*$ **then**
11:   $\mathcal{L}^* \leftarrow \mathcal{L}_t$, $z_{\text{poison}}^* \leftarrow p_{t-1}$
12:  **end if**
13:  $p_t \leftarrow \text{Jailbreak}(q, z_{\text{benign}}^{(k)}, s', p_{t-1}, \mathcal{L}_t)$      $\triangleright$ Next candidate
14: **end for**
15: **Return:** $z_{\text{poison}}^*$

---

observed standard deviations across seeds are as follows: Clean Accuracy (ACC)—2.32, Robust Accuracy (RACC)—3.78, and Attack Success Rate (ASR)—3.56.

### D.1. Adaptive Attacks

We extend existing jailbreak attacks such as GCG (Zou et al., 2023) and AutoDAN (Liu et al., 2023b) by optimizing a poisoned passage with full access to the query and retrieval context. Starting from an initial success from a prior attack, denoted as $z_{\text{poison}}$, we iteratively refine it to minimize a compute loss $\mathcal{L}_t$ that balances effectiveness and stealth.

The loss is defined as $\mathcal{L}_t = \mathcal{L}_1 + \lambda \cdot \mathcal{L}_2$, where $\mathcal{L}_1$ is the cross-entropy between between the model's response (given the corrupted retrieved set including $z_{\text{poison}}$) and the target answer $s'$, and $\mathcal{L}_2$ is the variance of the normalized attention scores over all passages in the retrieved set—encouraging low detectability. Here, $\lambda$ is a scalar parameter that balances the attack effectiveness with stealth.

At each iteration, we apply a jailbreak method, denoted as Jailbreak, to propose a modified candidate passage that minimizes $\mathcal{L}_t$. Among all generated candidates across iterations, we select the one yielding the lowest loss as the optimized poisoned passage. The full procedure is detailed in Algorithm 2.

In our experiments, we insert the poisoned passage at the last index of the retrieved set to construct the corrupted retrieved set. This placement eliminates retrieval randomness, enabling easier reproducibility and consistent comparison across queries—particularly important given the high computational cost of adaptive attacks. We also set the $\alpha = \infty$ for the AV Filter and select 20 queries from each dataset, prioritizing those where existing attacks were successful against Vanilla RAG. Since initialization from successful attacks typically yields a low value of $\mathcal{L}_1$, we terminate the optimization early if the attention variance $\mathcal{L}_2$ falls below the AV Filter threshold $\delta$. The attack is run for 100 steps using standard parameters for each jailbreak method.

We tune the scalar parameter $\lambda$ in the adaptive attack loss using the RealtimeQA dataset and Llama 2, evaluating values from the set $\{0.01, 0.1, 1\}$. We select $\lambda = 0.1$ for all subsequent adaptive attack experiments, as it yields the highest ASR.

Figure 4(a) represents the impact of varying $\lambda$ on attack performance. For evaluation, we apply adaptive attacks using jailbreak methods, including GCG and AutoDAN, initialized with poisoned passages generated by the PoisonedRAG attack (Poison). Table 7 reports the robust accuracy and attack success rate (RACC / ASR) of adaptive attacks against the AV Filter across multiple settings. The results show that adaptive attacks can potentially evade the AV Filter, achieving a maximum ASR of 35% and an average ASR of 22.08%.

*Table 7.* RACC and ASR of adaptive attacks (GCG and AutoDAN) initialized with poisoned passages from Poison against AV Filter, showing increased ASRs of up to 35%—higher than existing attacks on AV Filter but still lower than ASRs of Vanilla RAG and empirical upper bounds of other baselines.

| LLM | Adaptive Attack | RQA-MC | RQA | NQ |
|---|---|---|---|---|
| **Llama 2-C** | GCG-Poison | 55 / 15 | 35 / 30 | 15 / 10 |
| | AutoDAN-Poison | 35 / 35 | 40 / 20 | 25 / 10 |
| **Mistral-7B** | GCG-Poison | 50 / 25 | 25 / 25 | 35 / 35 |
| | AutoDAN-Poison | 50 / 20 | 20 / 15 | 30 / 25 |

Although adaptive attacks demonstrate reasonable success against the AV Filter, several limitations reduce the severity of the threat they pose. These attacks are highly dependent on the specific query, model, and benign retrieved set, requiring access to the LLM, the retriever, and the knowledge database—an assumption that may not hold for many practical adversaries. Furthermore, since adaptive attacks rely on iterative jailbreak methods, which are known for their high computational cost, they inherit long runtimes. Each poisoned passage must be individually optimized, significantly increasing the time required for the attack. Table 8 reports the average runtime per query (in seconds) across various settings, highlighting the computational overhead associated with these attacks. The AutoDAN-Poison attack on the RealtimeQA dataset using Mistral-7B incurred the highest average runtime among all settings, taking **18616.84** seconds per query. When executed sequentially on 20 queries, this resulted in a total runtime of approximately **4.3** days on a single H100 GPU. Figure 4(b) shows the loss trajectory for a randomly selected query from the RealtimeQA dataset during the adaptive attack on Llama 2.

*Table 8.* Average runtime of the adaptive attack per query across various settings. The runtime reaches up to $1.8 \times 10^4$ seconds, which is several orders of magnitude $\left(\sim \times 10^3\right)$ higher than the runtime of the existing attack Poison, as reported in (Zou et al., 2024).

| LLM | Adaptive Attack | RQA-MC | RQA | NQ |
|---|---|---|---|---|
| **Llama 2-C** | GCG-Poison | 7015.68 | 15833.36 | 9146.72 |
| | AutoDAN-Poison | 6233.20 | 18274.31 | 9737.39 |
| **Mistral-7B** | GCG-Poison | 8606.68 | 14890.24 | 9624.45 |
| | AutoDAN-Poison | 6604.01 | 18616.84 | 18248.52 |

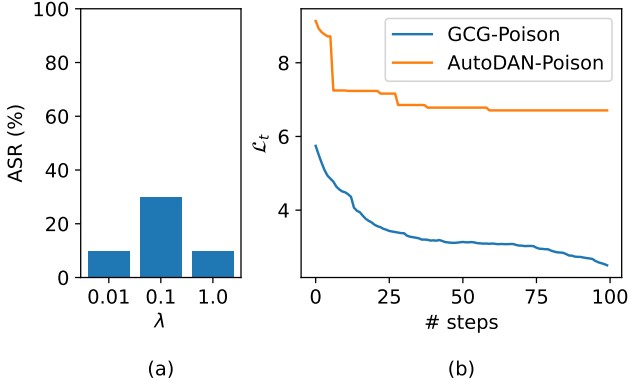

(a)        (b)

*Figure 4.* (a) Attack Success Rate (ASR) of the GCG-Poison adaptive attack on the RealtimeQA dataset using Llama 2 across varying values of $\lambda$, illustrating that $\lambda = 0.1$ achieves the highest ASR and is therefore selected for the rest of the evaluation. (b) Loss trajectory for a randomly selected query from RealtimeQA on Llama 2, demonstrates how the adaptive attack consistently reduces the target loss by lowering the variance of the corrupted retrieved set, thereby improving stealth.

*Table 9.* Robust Accuracy and Attack Success Rate (RACC/ASR) showing that AV Filter effectively mitigates additional content-poisoning attacks, even when they appear natural or semantically coherent to humans.

| LLM | Dataset Attack Defense | RQA-MC | | | RQA | | | NQ | | |
|---|---|---|---|---|---|---|---|---|---|---|
| | | Paradox (racc↑/asr↓) | MA (racc↑/asr↓) | Corrupt (racc↑/asr↓) | Paradox (racc↑/asr↓) | MA (racc↑/asr↓) | Corrupt (racc↑/asr↓) | Paradox (racc↑/asr↓) | MA (racc↑/asr↓) | Corrupt (racc↑/asr↓) |
| **Mistral-7B** | Vanilla | 54.2 / 41.4 | 60.4 / 31.8 | 52.0 / 32.0 | 30.4 / 35.0 | 39.4 / 26.6 | 43.6 / 17.8 | 29.2 / 24.2 | 56.8 / 5.00 | 40.4 / 15.8 |
| | $AV_{(\alpha=5)}$ | 76.0 / 8.40 | 74.0 / 8.60 | 69.4 / 10.8 | 57.6 / 4.20 | 54.0 / 8.00 | 57.2 / 7.20 | 50.6 / 7.40 | 53.2 / 4.40 | 48.4 / 9.20 |
| | $AV_{(\alpha=10)}$ | 77.4 / 6.80 | 73.6 / 9.00 | 68.2 / 11.0 | 59.4 / 4.80 | 54.0 / 7.80 | 59.2 / 8.00 | 52.0 / 7.60 | 54.8 / 3.80 | 51.0 / 9.20 |
| | $AV_{(\alpha=\infty)}$ | **80.0 / 3.40** | **77.2 / 5.20** | **74.6 / 7.60** | **65.2 / 3.80** | **65.6 / 2.80** | **61.8 / 7.40** | **56.4 / 4.00** | **56.2 / 2.00** | **51.2 / 8.00** |
| **Llama2-C** | Vanilla | 50.0 / 42.8 | 57.0 / 34.2 | 37.0 / 51.8 | 37.2 / 39.8 | 50.2 / 22.0 | 20.0 / 63.4 | 32.0 / 28.6 | 59.6 / 4.80 | 44.2 / 30.6 |
| | $AV_{(\alpha=5)}$ | 59.0 / 25.4 | 71.0 / 12.8 | 45.0 / 39.8 | 58.0 / 6.60 | 54.4 / 5.80 | 54.0 / 15.4 | 46.2 / 10.6 | 52.0 / 4.60 | 42.8 / 19.8 |
| | $AV_{(\alpha=10)}$ | 64.2 / 20.8 | 71.2 / 13.8 | 49.0 / 34.8 | 58.4 / 6.20 | 55.0 / 7.20 | 53.0 / 17.4 | 48.4 / 10.6 | 52.2 / 4.40 | 45.6 / 18.0 |
| | $AV_{(\alpha=\infty)}$ | **77.6 / 8.20** | **77.2 / 6.80** | **52.6 / 18.8** | **59.8 / 1.80** | **59.2 / 2.40** | **45.2 / 38.8** | **51.8 / 1.60** | **54.6 / 0.20** | **46.2 / 22.4** |
| **GPT-4o** | Vanilla | 31.6 / 37.2 | 41.0 / 25.0 | 53.4 / 4.60 | 41.2 / 45.0 | 48.4 / 33.0 | 61.2 / 11.6 | 37.4 / 22.4 | 59.4 / 2.80 | 56.8 / 2.40 |
| | $AV_{(\alpha=5)}$ | 56.8 / 5.60 | 50.4 / 11.0 | 54.2 / 3.60 | 63.2 / 7.60 | 57.4 / 9.40 | 66.0 / 6.80 | 54.0 / 6.80 | 57.0 / 2.40 | 56.8 / 6.00 |
| | $AV_{(\alpha=10)}$ | 58.0 / 3.80 | 51.0 / 10.2 | 54.8 / 3.20 | 65.2 / 6.00 | 58.0 / 8.60 | 65.6 / 6.00 | 54.6 / 6.60 | 58.0 / 2.00 | 57.0 / 5.40 |
| | $AV_{(\alpha=\infty)}$ | **62.8 / 2.40** | **59.6 / 4.20** | **55.2 / 3.00** | **66.4 / 4.60** | **68.2 / 2.40** | **65.2 / 5.20** | **57.8 / 3.20** | **60.8 / 0.00** | **59.8 / 3.40** |

*Table 10.* Detection Accuracy (DACC) of AV Filter against existing attacks, showing that AV Filter accurately removes the actual poisoned passage from the corrupted retrieved set, achieving the DACC up to 1.00 (perfect detection).

| Dataset LLM | top-$\alpha$ | RQA-MC | | RQA | | NQ | |
|---|---|---|---|---|---|---|---|
| | | PIA | Poison | PIA | Poison | PIA | Poison |
| **Mistral-7B** | $\alpha = 5$ | **1.00** | 0.88 | 0.99 | **0.97** | **1.00** | 0.67 |
| | $\alpha = 10$ | 0.99 | 0.89 | **1.00** | 0.93 | 0.99 | 0.69 |
| | $\alpha = \infty$ | 0.92 | **0.95** | 0.97 | 0.92 | 0.83 | **0.71** |
| **Llama2-C** | $\alpha = 5$ | 0.81 | 0.47 | **0.98** | 0.82 | 0.94 | 0.64 |
| | $\alpha = 10$ | **0.90** | 0.68 | **0.98** | **0.85** | **0.96** | 0.67 |
| | $\alpha = \infty$ | 0.88 | **0.77** | 0.94 | 0.79 | 0.88 | **0.70** |

### D.2. Additional Attacks

We also evaluate AV Filter on three additional content-poisoning attacks: Misinformation Attack (MA), Paradox, and CorruptRAG, as reported in Table 9. Results on other configurations are expected to follow similar trends. AV Filter remains effective against these attacks. On RQA-MC, it reduces the average attack success rate from $37.1\%$ with vanilla RAG to $9.7\%$, with comparable robustness across other datasets. Although content-poisoning attacks such as Poison, Paradox, MA, and CorruptRAG often appear natural and semantically coherent to humans, AV Filter detects them by analyzing LLM attention patterns rather than surface-level semantics, demonstrating that it does not rely on attack-specific semantic cues.

### D.3. AV Filter Detection Rate: Identifying Poisoned Passage

AV Filter is designed to identify and remove the potentially poisoned passages from a corrupted retrieved set, allowing the remaining (presumably benign) passages to be used for response generation. When the AV Filter successfully eliminates the actual poisoned passages, it is expected to improve the robust accuracy (RACC) and reduce the attack success rate (ASR)—a trend confirmed in our evaluation.

The consistent improvement in robustness over Vanilla RAG indicates that AV Filter reliably removes the correct poisoned passages. To explicitly quantify this behavior, we report the **Detection Accuracy (DACC)**—the fraction of successful attacks against Vanilla RAG in which AV Filter removes the actual poisoned passage. Table 10 presents the DACC across different $\alpha$ values used in the computation of NormScore$_\alpha$ and $\epsilon = 0.1$, demonstrating that AV Filter achieves high precision in removing the poisoned passage with an average detection accuracy of **0.86**. This reinforces AV Filter's effectiveness in accurately identifying and filtering poisoned passages from the retrieved set.

### D.4. AV Filter False Positive Rate

AV Filter estimates the influence of each passage in the retrieved set on the generated answer and, like other robust aggregators, assumes majority consensus: benign passages should agree on the correct answer and outnumber adversarial ones.

Even when the RAG pipeline returns the correct answer, some benign passages may receive disproportionately high attention scores and be removed. This is generally not a concern, as dropping a few benign passages from a largely benign set rarely affects the output. As shown in Table 2 and 13, the accuracy drop for benign retrievals is limited to **4–6%**, substantially smaller than for other baselines.

We also report the False Positive Rate (FPR) of AV Filter ($\alpha = \infty$) for $\delta \in \{10, 26.2, 30, 40\}$ on benign retrievals (Table 11), with similar trends expected across other configurations. Any removal of a passage from a benign set is counted as a false positive. For corrupted sets, ASR provides a reasonable upper bound for FPR. For RQ2 and RQ3, we adopt $\delta = 26.2$ as the evaluation setting.

*Table 11.* False Positive Rate (FPR) of AV Filter on benign retrievals. The average FPR is 0.24. We allow a slightly higher rate, as removing a few benign passages is less harmful than retaining an adversarial one, which could compromise the output.

| LLM | Dataset | $\delta = 40$ | $\delta = 30$ | $\delta = 26.2$ | $\delta = 10$ |
|---|---|---|---|---|---|
| **Mistral-7B** | **RQA-MC** | 0.09 | 0.11 | 0.11 | 0.18 |
| | **RQA** | 0.22 | 0.26 | 0.26 | 0.33 |
| | **NQ** | 0.27 | 0.33 | 0.36 | 0.41 |
| **Llama2-C** | **RQA-MC** | 0.05 | 0.06 | 0.09 | 0.21 |
| | **RQA** | 0.15 | 0.20 | 0.24 | 0.38 |
| | **NQ** | 0.24 | 0.29 | 0.36 | 0.45 |

### D.5. Additional Baseline Defense Strategies

We compare AV Filter with several baseline defenses, which often suffer from high false positive rates:

(i) **Perplexity Filtering:** The same model as the RAG LLM computes the perplexity of each passage (Mistral-7B is used for GPT-4o). The passage with the highest perplexity is removed, under the heuristic that it may be maliciously generated.

(ii) **Vigilant Prompting:** A defensive prompting strategy that warns the LLM about possible misinformation. For example, QA prompts include cautions such as: *"Be aware that some passages may be designed to mislead you."*

(iii) **Reranking Methods:** Separate models rerank retrieved passages by relevance to the query. For comparison, we use transformer-based models (ColBERTv2 and T5 seq2seq). Notably, the typical use of rerankers—passing the most relevant passages to the model—is not adversarially robust. Poisoned passages consistently receive high relevance scores because they contain a clear answer to the query, albeit an incorrect one. Since rerankers are not trained to detect adversarial corruption, they inherently treat poisoned passages as highly relevant. For a fair comparison, we therefore remove the passage ranked highest in relevance, as it is most likely to be poisoned. This is not the typical use of rerankers but provides a stronger baseline than standard relevance-based reranking, which offers no defense against adversarial attacks.

Table 12 reports the attack success rates of these baselines against Poison, PIA, and Paradox, compared with AV Filter ($\alpha = \infty$) under the RQ2 setup, with similar trends expected across other configurations.

### D.6. Wikipedia Corpus

We evaluate AV Filter against existing attacks using the Wikipedia Corpus as the Knowledge database, demonstrating its effectiveness across varying knowledge distributions. Specifically, we use 100 queries each from the HotpotQA and NQ datasets, retrieving top 10 passages from the Wikipedia corpus using the Contriver retriever. We utilize the Wikipedia corpus and retrieval results publicly released by PoisonedRAG (Zou et al., 2024).

Similar to our evaluation with Google Search as the knowledge database, we report the Clean Accuracy (ACC), Robust Accuracy (RACC), and Attack Success Rate (ASR) on the HotpotQA and NQ datasets using the Wikipedia corpus as the underlying knowledge base.

Table 13 reports the clean accuracy across different models, datasets, and values of $\alpha$. The AV Filter preserves high clean performance, with only a modest average drop of $4 - 6\%$ across datasets, with similar trends expected across other configurations.

Table 14 presents the Robust Accuracy (RACC) and Attack Success Rate (ASR) achieved by AV Filter for varying values

*Table 12.* ASR shows that AV Filter outperforms other defenses in 6/9 Poison cases, 9/9 Paradox cases, and 1/9 PIA cases. Performance on PIA is lower because PIA embeds verbatim query in poisoned passages, which makes them especially easy for reranking methods to detect.

| | Dataset | RQA-MC | | | RQA | | | NQ | | |
|---|---|---|---|---|---|---|---|---|---|---|
| **LLM** | **Attack** | **PIA** | **Poison** | **Paradox** | **PIA** | **Poison** | **Paradox** | **PIA** | **Poison** | **Paradox** |
| | **Defense** | | | | | | | | | |
| **Mistral-7B** | Perplexity Filter | 17.6 | 31.6 | 55.2 | 14.4 | 25.0 | 31.8 | 3.60 | 11.4 | 27.6 |
| | Vigilant Prompt | 32.2 | 28.6 | 54.2 | 27.4 | 21.0 | 27.8 | 21.6 | 9.20 | 20.2 |
| | Reranking (ColBERTv2) | **3.00** | 10.0 | 14.0 | **2.00** | **5.00** | 8.00 | **2.00** | 5.00 | 12.0 |
| | Reranking (t5) | 5.00 | 15.0 | 12.0 | 2.40 | 8.60 | 7.00 | 6.00 | 7.00 | 13.0 |
| | **AV Filter**$_{(\alpha=\infty)}$ | 7.20 | **8.40** | **3.40** | 2.40 | 8.60 | **3.80** | 5.80 | **4.00** | **4.00** |
| **Llama2-C** | Perplexity Filter | 34.8 | 28.8 | 43.8 | 42.0 | 17.6 | 41.6 | 6.40 | 7.40 | 33.0 |
| | Vigilant Prompt | 64.0 | 29.4 | 49.8 | 89.6 | 16.4 | 36.2 | 76.0 | 7.20 | 28.6 |
| | Reranking (ColBERTv2) | **6.00** | 13.0 | 15.0 | **4.00** | 9.00 | 15.0 | 14.0 | 4.00 | 13.0 |
| | Reranking (t5) | 18.0 | 17.0 | 16.0 | 21.0 | 10.0 | 14.0 | 31.0 | 6.00 | 19.0 |
| | **AV Filter**$_{(\alpha=\infty)}$ | 16.8 | **12.6** | **8.20** | 5.00 | **6.60** | **1.80** | 9.20 | **3.60** | **1.60** |
| **GPT-4o** | Perplexity Filter | 7.60 | 23.2 | 37.0 | 15.0 | 28.4 | 47.2 | 1.80 | 7.20 | 24.4 |
| | Vigilant Prompt | 16.2 | 23.6 | 34.6 | 15.0 | 23.8 | 38.2 | 10.8 | 5.60 | 14.8 |
| | Reranking (ColBERTv2) | **0.40** | **6.00** | 4.80 | **1.00** | 10.6 | 10.0 | 5.20 | **1.00** | 8.00 |
| | Reranking (t5) | 2.00 | 7.20 | 4.60 | 7.00 | 11.2 | 10.6 | 10.6 | 2.00 | 9.00 |
| | **AV Filter**$_{(\alpha=\infty)}$ | 5.40 | 6.80 | **2.40** | 7.00 | **9.20** | **4.60** | 11.4 | 1.60 | **3.20** |

*Table 13.* Clean Accuracy (ACC) of defenses, showing that AV Filter preserves RAG utility with a minimal drop from Vanilla, achieving up to 10% higher ACC than other baseline defenses.

| **LLM** | **Mistral-7B** | | **Llama2-C** | | **GPT-4o** | |
|---|---|---|---|---|---|---|
| **Defense** | **HotpotQA** | **NQ** | **HotpotQA** | **NQ** | **HotpotQA** | **NQ** |
| Vanilla | 51.0 | 59.0 | 36.0 | 46.0 | 45.6 | 47.4 |
| Keyword | **59.0** | 49.0 | **43.0** | 37.0 | 44.6 | **55.0** |
| Decoding | 41.0 | 50.0 | 26.0 | 28.0 | – | – |
| **AV Filter**$_{(\alpha=5)}$ | 40.0 | 43.0 | 27.0 | 34.0 | **45.0** | 48.2 |
| **AV Filter**$_{(\alpha=10)}$ | 46.0 | 44.0 | 27.0 | 36.0 | 44.8 | 47.4 |
| **AV Filter**$_{(\alpha=\infty)}$ | 51.0 | **59.0** | 36.0 | **46.0** | 44.2 | 47.6 |

of $\alpha$ used in computing $\mathsf{NormScore}_\alpha$. The results demonstrate that the AV Filter often outperforms baseline defenses in robustness, achieving up to **9.8**% higher RACC, with similar trends expected across other configurations. Furthermore, even at a low corruption rate of $\epsilon = 0.1$, Vanilla RAG remains highly vulnerable, with ASR reaching up to **90.2**%. In contrast, AV Filter significantly reduces this vulnerability—bringing the average ASR down to **15.36**% on the HotpotQA and **14.71**% on the NQ dataset.

Notably, the ASR for the Keyword and Decoding defenses is anomalously high on the HotpotQA dataset. This is attributed to the multi-hop nature of many HotpotQA queries, which often require reasoning across multiple passages. Since both variants of Certified Robust RAG evaluate each passage in isolation, they fail to aggregate information across passages to answer correctly. As a result, they are more susceptible to a single poisoned passage that contains complete information aligned with the adversarial target answer.

### D.7. Combining AV Filter with Other Defenses

As a detection-based pruning defense, AV Filter can be used as a preprocessing step alongside other strategies, such as Certified Robust RAG, to further reduce attack success rates. However, the robust accuracy of the ensemble may still be limited by the underlying defense.

We combine AV Filter ($\alpha = \infty$) with Certified Robust RAG-Keyword by first removing potentially poisoned passages using AV Filter and then applying Keyword for robust generation. Table 15 reports the robust accuracy and attack success rates,

*Table 14.* Robust Accuracy and Attack Success Rate (RACC/ASR) showing that AV Filter effectively mitigates attacks with low ASRs while achieving up to 9.8% higher RACC than baselined defenses.

| | Dataset | HotpotQA | | NQ | |
|---|---|---|---|---|---|
| **LLM** | **Attack** | **PIA** | **Poison** | **PIA** | **Poison** |
| | **Defense** | (racc↑ / asr↓) | (racc↑ / asr↓) | (racc↑ / asr↓) | (racc↑ / asr↓) |
| **Mistral-7B** | Vanilla | 18.6 / 69.0 | 14.6 / 75.0 | 22.2 / 55.8 | 23.0 / 50.4 |
| | Keyword | 48.0 / 21.0 | 43.0 / 25.0 | 40.0 / 7.0 | 42.0 / **10.0** |
| | Decoding | 38.0 / 28.0 | 30.0 / 51.0 | 47.0 / 7.0 | 43.0 / 20.0 |
| | **AV Filter**$_{(\alpha=5)}$ | **53.0 / 8.0** | 47.4 / 14.8 | **49.8 / 11.0** | 43.0 / 14.6 |
| | **AV Filter**$_{(\alpha=10)}$ | 52.6 / 8.4 | 47.8 / 15.0 | 48.6 / 11.2 | **44.0** / 13.2 |
| | **AV Filter**$_{(\alpha=\infty)}$ | 47.2 / 13.4 | **48.8 / 13.6** | 36.6 / 26.8 | 42.2 / 12.4 |
| **Llama 2-C** | Vanilla | 3.6 / 90.2 | 14.6 / 65.6 | 6.4 / 85.6 | 26.2 / 48.0 |
| | Keyword | **36.0** / 25.0 | **41.0** / 20.0 | 36.0 / 8.0 | 37.0 / **9.0** |
| | Decoding | 23.0 / 33.0 | 25.0 / **16.0** | 24.0 / 30.0 | 26.0 / 23.0 |
| | **AV Filter**$_{(\alpha=5)}$ | 34.0 / 11.4 | 27.0 / 17.0 | 42.6 / 6.4 | **37.2** / 17.6 |
| | **AV Filter**$_{(\alpha=10)}$ | 34.4 / **10.4** | 26.8 / 17.0 | **44.2 / 6.2** | 36.4 / 15.6 |
| | **AV Filter**$_{(\alpha=\infty)}$ | 17.8 / 44.0 | 21.4 / 28.6 | 22.4 / 45.6 | 32.4 / 25.8 |
| **GPT-4o** | Vanilla | 10.6 / 78.8 | 20.4 / 58.4 | 16.8 / 69.4 | 28.6 / 34.6 |
| | Keyword | **43.6** / 17.4 | **43.4** / 15.8 | **53.2** / 6.2 | **53.0** / 4.8 |
| | **AV Filter**$_{(\alpha=5)}$ | 42.6 / **9.8** | 37.2 / 12.6 | 40.2 / **5.8** | 36.8 / 6.8 |
| | **AV Filter**$_{(\alpha=10)}$ | 40.0 / 11.2 | 37.6 / 12.0 | 41.6 / 6.8 | 35.8 / **4.0** |
| | **AV Filter**$_{(\alpha=\infty)}$ | 35.6 / 17.8 | 41.2 / **11.4** | 28.0 / 29.6 | 38.6 / 5.4 |

*Table 15.* Robust accuracy and attack success rates for the combined defense. The combination consistently outperforms individual defenses, reducing attack success rates to an average of **1.22**% across all cases.

| | Dataset | RQA-MC | | RQA | | NQ | |
|---|---|---|---|---|---|---|---|
| **LLM** | **Attack** | **PIA** | **Poison** | **PIA** | **Poison** | **PIA** | **Poison** |
| | **Defense** | (racc↑ / asr↓) | (racc↑ / asr↓) | (racc↑ / asr↓) | (racc↑ / asr↓) | (racc↑ / asr↓) | (racc↑ / asr↓) |
| **Mistral-7B** | Keyword | 57 / 7 | 56 / 6 | 54 / 6 | 55 / 6 | 50 / 1 | **53** / 5 |
| | AV Filter$_{(\alpha=\infty)}$ | **79** / 6 | **73** / 8 | **62** / 6 | 54 / 6 | 53 / 7 | 52 / 4 |
| | Keyword + AV Filter$_{(\alpha=\infty)}$ | 58 / 3 | 58 / 3 | 60 / 3 | **60 / 3** | **53 / 0** | **53 / 0** |
| **Llama2-C** | Keyword | 53 / 6 | 55 / 6 | 53 / 6 | 55 / 6 | 52 / 2 | 53 / 4 |
| | AV Filter$_{(\alpha=\infty)}$ | **70** / 18 | **71** / 13 | **61** / 2 | 56 / 6 | **54** / 4 | **54** / 5 |
| | Keyword + AV Filter$_{(\alpha=\infty)}$ | 58 / **0** | 58 / **0** | 58 / **0** | **58 / 0** | 53 / 1 | 53 / **1** |
| **GPT-4o** | Keyword | 62 / 13 | 60 / 17 | 63 / 15 | **63** / 15 | 58 / 8 | 61 / 6 |
| | AV Filter$_{(\alpha=\infty)}$ | 59 / 4 | 50 / 5 | **69** / 2 | 59 / 9 | 59 / 1 | **62** / 1 |
| | Keyword + AV Filter$_{(\alpha=\infty)}$ | **64 / 2** | **63 / 2** | 63 / 2 | 63 / 2 | **62 / 0** | **62 / 0** |

with similar trends expected across other configurations. The combined defense consistently achieves lower attack success rates than either method alone, with an average of just **1.22**% across all cases.

### D.8. Hyperparameter Analysis

**Corruption Fraction** $\epsilon$. We evaluate the AV Filter under varying corruption fractions to its robustness as the rate of corruption increases. Specifically, we measure Robust Accuracy (RACC) and Attack Success Rate (ASR) on the RealtimeQA-MC dataset across multiple models, using a fixed $\alpha = \infty$ and a single random seed. Figure 5(a) and (b) present the average RACC and ASR for corruption rates $\epsilon \in \{0.1, 0.2, 0.3, 0.4\}$, with the total retrieved set size fixed at $k = 10$. As expected, increasing the corruption fraction leads to higher ASR and lower RACC. Nevertheless, the AV Filter remains reasonably effective even under high corruption—reducing ASR to 32.67% at $\epsilon = 0.4$ for Poison. We expect a similar trend for other datasets and $\alpha$ values.

**Filtering Threshold** $\delta$. The effectiveness of the AV Filter depends on the filtering threshold $\delta$, which governs the acceptable variance in attention score across the retrieved set. We set $\delta = 26.2$ for our main experiments, estimated from clean retrievals on the RealtimeQA dataset using Llama 2. This estimated threshold generalizes well, as it yields strong performance across

different datasets and models. To further assess the robustness of the AV Filter to this hyperparameter, we evaluate its performance across a range of thresholds $\delta \in \{10, 20, 30, 40\}$. Specifically, we report Robust Accuracy (RACC) and Attack Success Rate (ASR) on the RealtimeQA-MC dataset, averaged over multiple models using a fixed $\alpha = \infty$ and a single random seed. Figure 5(c) and (d) show that both RACC and ASR remain relatively stable across this range, indicating that AV Filter is not overly sensitive to $\delta$ and can generalize well to unseen data without requiring fine-tuning. We expect a similar trend for other datasets, attacks, and $\alpha$ values.

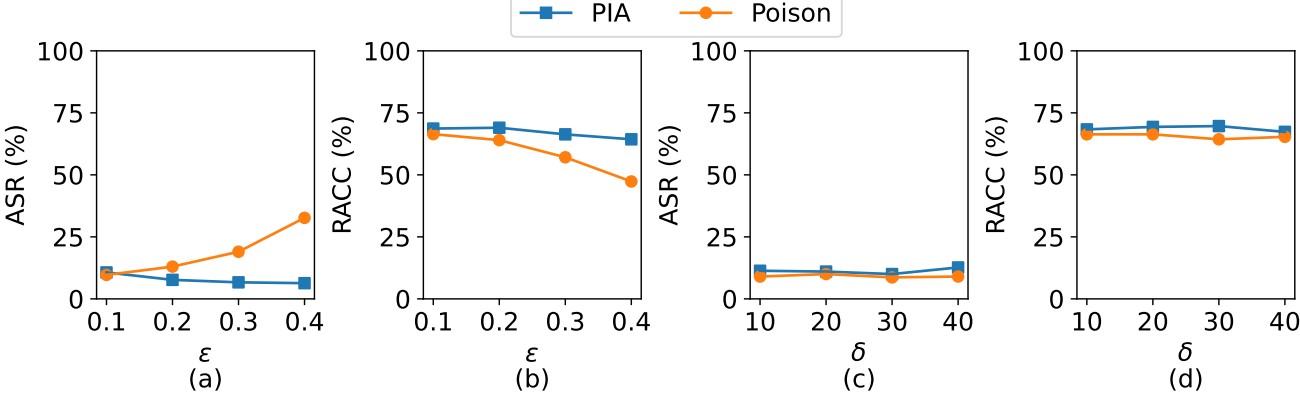

*Figure 5.* **Effect of Corruption Rate and Filtering Threshold:** This figure shows the impact of varying the corruption rate $\epsilon$ and the filtering threshold $\delta$ on the performance of the AV Filter. Subfigures (a) and (b) present the Attack Success Rate (ASR) and Robust Accuracy (RACC) on the RealtimeQA-MC dataset with $\alpha = \infty$, averaged over all models. As expected, ASR increases, and RACC decreases with higher corruption rates. Subfigures (c) and (d) report ASR and RACC for varying $\delta$ values, again averaged over all models, demonstrating that AV Filter's performance is not overly sensitive to the threshold. This indicates that AV Filter can generalize well to unseen data without requiring fine-tuning of $\delta$.

# E. LLM Usage

We did not make use of LLMs in the writing or research process beyond minor revisions to the text.

