# OpenReview forum: "Through the Stealth Lens: Attention-Aware Defenses Against Poisoning in RAG"
_ICML.cc/2026/Conference — ICML 2026 regular_

### Official Review · Reviewer_zSA2 · 2026-03-13

**Soundness:** 3
**Presentation:** 3
**Significance:** 2
**Originality:** 3
**Overall Recommendation:** 4
**Confidence:** 3

**Summary:**

This paper studies the stealth of poisoning attacks against Retrieval-Augmented Generation (RAG) systems. The key insight was that in order for the poisoned passage to heavily influence the victim models, the attention on those poisoned passages should be abnormally high. Based on this observation, they designed an attention variance filter (AV Filter) that removes passages with abnormally high attention scores. The results show that the AV Filter robustly detected the poison passages, outperforming the baseline. They also attempted adaptive attacks that are very effective but less practical due to high computation cost.

**Compliance With Llm Reviewing Policy:**

Affirmed.

**Final Justification:**

Since all of my concerns are resolved, I'm going to keep my positive assessment.

**Key Questions For Authors:**

1. Have you validated that NPAS actually correlates with causal influence on the output? A leave-one-out comparison (removing one passage at a time and measuring output change) against NPAS would validate the core assumption.

2. Given that the false positive rate varies (as shown in Table 11) with the setting of delta, what would be a reasonable way to calibrate delta and set a suitable value, especially when testing on different datasets with different models?

**Limitations:**

Yes

**Strengths And Weaknesses:**

Strength

1. Interesting key observation

The key observation that the poisoned passages demonstrate abnormally high attention scores is interesting and seems to be effective.

2 Lightweight Design

The AV filter does not require additional training and does not put too much computation overhead to the inference with RAG

3 Adaptive Attacks

 Rather than claiming the defense is perfect, the authors tried adaptive attacks that can partially bypass the AV Filter.

Weaknesses

1. Attention filter Is Not Well-Validated

The paper's proposed method relies on the assumption that the attention scores can reliably indicate whether or not the passage is actually poisoned. This claim has not been reliably validated in the paper; hence, it makes the claims less convincing.

2. Threshold Sensitivity and Generalization

The filtering threshold delta is estimated from a single dataset and a single model. Different domains, passage lengths, or LLM architectures might require different thresholds. Since the false positive rate seems to be affected by this threshold, it makes more sense to have a deeper analysis into how to find a suitable delta.

---

> ### Author Rebuttal · Authors · 2026-03-29
>
> We thank the reviewer for their thoughtful feedback and for highlighting the key strengths of our work. We clarify the remaining concerns and questions through our response below.
>
> &nbsp;
>
> ### W1: Attention Filter Not Well-Validated
>  By construction of the benchmarks, adversarial answers are chosen at random and appear only in the poisoned passage. Therefore, when the LLM produces an adversarial response over multiple seeds, the poisoned passage is necessarily the cause. This is reflected in the gap between Clean Accuracy and Robust Accuracy (Tables 2 and 3), where the sole difference between the two settings is the presence of a single poisoned passage, which reduces RACC and increases the ASR.
>
> We validate that high NPAS correlates with the high influence of poisoned passages of existing attacks through Detection Accuracy (DACC). DACC measures, among all instances where the poisoned passage influenced Vanilla RAG to produce the adversarial response, the fraction where the highest NPAS is in fact the poisoned one. As reported in Table 10 (for $\epsilon = 0.1$ and $k = 10$), AV Filter achieves an average DACC of $0.86$ across configurations. We provide an extended DACC table below.
>
> |LLM\Dataset|α|RQA-MC||RQA||NQ||
> |:-:|:-:|:-:|:-:|:-:|:-:|:-:|:-:|
> |||PIA|Poison|PIA|Poison|PIA|Poison|
> |Mistral-7B|5|0.96|0.93|0.95|0.96|0.86|0.74|
> ||10|0.95|0.96|0.95|0.96|0.84|0.74|
> ||∞|0.92|0.99|0.90|1.00|0.76|0.82|
> |Llama2-C|5|0.72|0.53|1.00|0.95|0.99|0.87|
> ||10|0.78|0.69|0.99|0.95|0.99|0.87|
> ||∞|0.79|0.88|0.98|0.97|0.93|0.87|
> |Llama-3.1|5|0.92|0.70|0.99|0.90|0.97|0.65|
> ||10|0.92|0.70|0.97|0.89|0.94|0.69|
> ||∞|0.91|0.80|0.97|0.91|0.93|0.72|
> |Deepseek-R1|5|0.86|0.80|0.33|0.22|0.27|0.36|
> ||10|0.86|0.80|0.52|0.32|0.38|0.44|
> ||∞|0.71|0.83|0.67|0.56|0.48|0.60|
>
> The Attack Success Rate (ASR) analysis is analogous to the leave-one-out causal validation for our experimental settings. In Table 3 ($\epsilon = 0.1$ and $k=10$), only a single poisoned passage is present in the retrieved set, causing the LLM to output the adversarial response by the design of the benchmark. AV Filter in this setup removes at most one passage, specifically the one with the highest NPAS. The substantial drop in ASR (for example, from 88.2% to 6.6% on RQA-MC) after the removal of a single passage by AV Filter confirms that the passage removed was causally responsible for the adversarial response. This is precisely a leave-one-out comparison: removing a single passage by the AV Filter changes the output from the adversarial answer to correct one, and ASR reduction captures the magnitude of this change. We are willing to do any specific leave-one-out analysis as suggested by the reviewer to make this point clearer.
>
> &nbsp;
>
> ### W2: Threshold Sensitivity and Generalization
>
> We report the variation across configurations due to a fixed filtering threshold in Table 11 through the false positive rate (FPR), measured on benign retrievals where any removal of a passage counts as false positive. At our chosen $\delta = 26.2$, FPR ranges from $0.09$ to $0.36$ across models and datasets, with an average of $0.24$. While we observe this per-configuration variation in FPR, it does not manifest in robust accuracy (RACC) or attack success rate (ASR). An aggressive threshold like ours will successfully filter poisoned passages as long as they exhibit high NPAS, which existing attacks consistently do, at the cost of occasionally removing benign passages. Due to redundancy in the retrieved set, moderate false positive rates have limited impact on accuracy, as reflected by 4-6% drop across settings (Tables 2 and 13).
>
> Since NPAS is normalized, the underlying problem for AV Filter across all configurations reduces to: given $k$ points in $[0, 100]$ summing to $100$, identify and remove up to $\epsilon \cdot k$ outliers. We acknowledge that even after normalization, scores are still model/dataset specific, but the variation is potentially limited by mapping raw attention weights from different configurations to the same scale.
>
> Finally, our choice to calibrate the filtering threshold on a single randomly chosen configuration (Llama 2, RQA) was deliberate: we wanted to demonstrate that we are not tuning per-configuration to get favorable numbers. The consistent ASR reductions (Table 2, 3, 13, 14) support this. In practice, if a representative test set is available, we recommend selecting an acceptable clean accuracy drop and calibrating $\delta$ by measuring the FPR. We will surface the sensitivity analysis (Table 11, Appendix D.8) more prominently in the revision.
>
> **We hope our response clarifies the remaining concerns. In light of these clarifications, we would kindly request the reviewer to consider increasing their rating. We are happy to provide further explanation or any specific experimental evidence on any point.**

---

> > ### Author Rebuttal · Reviewer_zSA2 · 2026-04-02
> >
> > Thank you for addressing my concerns. I will keep my positive rating.

---

> > > ### Author Response · Authors · 2026-04-04
> > >
> > > We thank the reviewer for engaging with our rebuttal and for confirming that all concerns have been resolved. We appreciate the positive rating and are glad our responses were helpful. If there is any further suggestion that would help move toward a stronger recommendation, we would be happy to address it.

---

### Official Review · Reviewer_7yF7 · 2026-03-13

**Soundness:** 3
**Presentation:** 3
**Significance:** 3
**Originality:** 3
**Overall Recommendation:** 4
**Confidence:** 3

**Summary:**

This paper argues that existing low-corruption poisoning attacks on retrieval-augmented generation are not truly stealthy. The authors formalize stealth via a distinguishability game (SADG), propose the Normalized Passage Attention Score (NPAS) as a proxy for passage influence on the final response, and build an Attention-Variance Filter (AV Filter) that removes high-influence outlier passages. The paper evaluates across multiple datasets, LLMs, and attacks, and additionally develops adaptive attacks that jointly optimize attack success and reduced attention variance. The main empirical claim is that AV Filter substantially lowers attack success rate while preserving clean accuracy better than several baselines.

**Compliance With Llm Reviewing Policy:**

Affirmed.

**Final Justification:**

My concerns have been addressed

**Key Questions For Authors:**

See *limitations*

**Limitations:**

My first concern is conceptual. The defense relies on attention weights as a proxy for causal influence. This is a reasonable engineering choice, but the paper occasionally speaks as though high attention variance itself explains or certifies corruption. That is too strong. Attention is not causality, and attention-based signals can vary across architectures, prompting schemes, output lengths, and decoding settings. I would like the final version to state this limitation more carefully and separate the theoretical intuition from what is actually measured.

Second, the thresholding strategy seems under-justified. The manuscript states that the threshold δ is estimated from benign RQA with Llama 2 at α = ∞ and then used more broadly. That is surprisingly global. A defense that depends on a threshold learned from one model/dataset pair may be brittle under distribution shift. Since the paper’s thesis is partly about generality, I would like to see either a sensitivity analysis in the main paper or a stronger argument for threshold transferability.

Third, the method may depend on white-box or semi-white-box access more than the title suggests. The paper does show a black-box variant using an auxiliary open-source model, which is good, but I would like clearer quantification of how much performance degrades under true transfer. If the auxiliary model differs substantially from the target generator, does NPAS remain predictive? This matters for deployment against closed models.

Fourth, the adaptive attack evaluation is informative but limited. The paper evaluates only 20 queries per dataset due to computational constraints. I understand the cost, but this small sample makes it harder to tell whether the observed stealth-effectiveness trade-off is stable. It would help if the authors could report variance bars or more detail on failure modes.

Fifth, I am not fully convinced the comparison to certified robust RAG baselines is always apples-to-apples. The paper argues, plausibly, that isolate-then-aggregate baselines suffer on multi-hop tasks, but that is also partly a difference in defense design goals. The framing should be slightly more careful in distinguishing “better under these tasks and metrics” from “strictly dominates.”

**Strengths And Weaknesses:**

First, the problem formulation is clear and well motivated. Many RAG poisoning papers implicitly treat stealth as “the poisoned passage looks plausible,” whereas this paper defines stealth in terms of distinguishability under the model’s actual behavior. That is a meaningful conceptual step.

Second, NPAS and AV Filter are simple, interpretable, and deployable. The method does not require retraining the generator or retriever, and the algorithmic design is easy to understand.

Third, the empirical evaluation is broad. The paper tests multiple attacks, datasets, and LLMs, and reports both clean accuracy and robust accuracy / ASR. The finding that AV Filter often preserves utility better than baselines while sharply reducing ASR is compelling.

Fourth, the adaptive attack section improves the paper considerably. It would have been easy to publish a defense against straw-man attacks; instead, the authors show that adaptive attacks can partially evade the filter, but only with stronger assumptions and nontrivial per-query optimization cost. That is exactly the kind of result I like: messy, limited, and therefore believable.

---

> ### Author Rebuttal · Authors · 2026-03-30
>
> We thank the reviewer for their thoughtful feedback, highlighting key strengths of our work, and appreciating our effort. We clarify the remaining concerns and questions below.
>
> &nbsp;
>
> ### W1: Attention Weights as a Proxy for Causal Influence
> For the existing attacks and configurations we evaluate, NPAS does correlate strongly with the causal influence of poisoned passages, as validated by the high detection accuracy (DACC of 0.86 on average, Table 10 and Reviewer zSA2 W1) and the substantial ASR reductions after filtering.
>
> That said, we acknowledge that this correlation may not hold universally across all architectures or decoding strategies, and we may need better proxies for influence in those settings. We do not claim that high attention variance is a necessary condition for successful poisoning; our adaptive attack evaluation also highlights this, showing that attacks can partially reduce their attention signature. We will revise the “Why analyze attention weights?” paragraph in Section 4 to state this limitation more carefully and add it to the limitations section as well.
>
> &nbsp;
>
> ### W2: Thresholding Strategy
> We note that while FPR (rate of removing a passage from clean retrieved set) varies across configuration (Table 11) due to fixed $\delta$, this variation does not manifest in RACC or ASR. _For more details, please refer to W2 in the response of Reviewer zSA2_. We will surface the sensitivity analysis (Table 11, Appendix D.8) more prominently in the revision.
>
> &nbsp;
>
> ### W3: White-box vs Black-box Transfer
> We want to clarify what the black-box setup is actually doing, as we think the concern may stem from a misunderstanding of the transfer mechanism. In our setup for GPT-4o, the auxiliary model (Mistral-7B) independently processes the same retrieved passages and query, computes its own NPAS, and identifies outlier passages. The filtering happens before the passages are sent to the target generator. The question is whether a poisoned passage designed to exert disproportionate influence produces a skewed attention signature in the auxiliary model as well.
>
> This holds for the existing attacks and models we evaluated, and the GPT-4o results in Table 3 confirm this empirically: the ASR reduction and RACC improvement are comparable to the white-box models. We agree that there will be gaps in the generative abilities between the closed-source target generator and the auxiliary model, mostly in that it may be harder to poison the more capable target generator, as our experiments also suggest. Still, this pre-processing setup creates a dilemma for the attacker, they must craft passages influential enough to steer the more robust target generator, while not being so aggressive that they leave detectable traces in the weaker auxiliary model. In fact, a defender could further exploit this asymmetry by training a purposefully vulnerable auxiliary model that amplifies the attention signature of poisoned passages, making detection even easier. We will discuss this trade-off more explicitly in the revision.
>
> &nbsp;
>
> ### W4: Adaptive Attack Eval
> We evaluated 240 queries, each over 3 choices of $\lambda$, using a fixed random seed. A single configuration takes 4 days, limiting our ability to expand this evaluation during the rebuttal. We will prioritize adding more results in the revision.
>
> That said, the current results provide valuable insights into the difficulty of optimizing attacks against AV Filter. We select queries where the attack succeeded on Vanilla RAG but failed with AV Filter, isolating the effort to minimize attention variance. The loss function $L_1 + \lambda \cdot L_2$, where $L_1$ is the target answer loss and $L_2$ is the attention variance, reveals a fundamental tension in the optimization. The attacks starts with low $L_1$ but high $L_2$, the heavy hitters contributing most to NPAS are precisely the tokens semantically related to or part of the target answer (Figure 3). With high $\lambda$, the optimization tries to modify these tokens to reduce variance, but doing so disrupts the answer and increases $L_1$, often forcing the optimization to revert these changes in subsequent steps. With calibrated $\lambda$, the optimization reduces variance through other tokens, but answer-bearing tokens remain the dominant contributors to NPAS.
>
> This trade-off reflects a real tension: the very tokens that make the attack successful are the ones that make it detectable. A defender aware of adaptive attacks could also calibrate $\delta$ more aggressively, further constraining the attacker.
>
> &nbsp;
>
> ### W5: Comparison to Certified Robust RAG
> We agree with the reviewer and will frame this comparison more carefully in the revision. _For more details, please refer to W2 in the response of Reviewer CpFQ_.
>
> &nbsp;
>
> **In light of these clarifications, we would kindly request the reviewer to consider increasing their rating and champion our work. We are happy to provide further explanation on any point.**

---

> > ### Author Rebuttal · Reviewer_7yF7 · 2026-04-02
> >
> > My concerns have been addressed
> >
> > I will keep my positive rating

---

> > > ### Author Response · Authors · 2026-04-04
> > >
> > > We thank the reviewer for engaging with our rebuttal and for confirming that all concerns have been resolved. We appreciate the positive rating and are glad our responses were helpful. If there is any further suggestion that would help move toward a stronger recommendation, we would be happy to address it.

---

### Official Review · Reviewer_CpFQ · 2026-03-14

**Soundness:** 3
**Presentation:** 3
**Significance:** 3
**Originality:** 2
**Overall Recommendation:** 4
**Confidence:** 3

**Summary:**

This paper proposes an attention-based defense for poisoning attacks in retrieval-augmented generation systems. The idea is to use normalized passage attention scores to measure how much each retrieved passage influences the model’s response, and then filter out anomalous passages with unusually concentrated attention. Experiments show the effectiveness of the proposed method.

**Compliance With Llm Reviewing Policy:**

Affirmed.

**Key Questions For Authors:**

see weaknesses

**Limitations:**

yes

**Strengths And Weaknesses:**

Strengths

* The investigated problem is interesting. The security of RAG system is important.

* This paper is well-written.

* The performance of the proposed method is good, achieving up to 20% higher accuracy than baselines.

* The adaptive attack is discussed and involved in the experiments.

Weaknesses

* The idea of detecting poisoned or adversarial samples through outlier signals in activations, features, or attention is not particularly new. Similar directions have been explored extensively since the CNN era (such as Chen et al. and Hayase et al.), so the main contribution here feels more like an modification and extension of this general line of defense to the LLM and RAG poisoning setting, which somewhat weakens the novelty of the paper.

* Although the overall results are strong, they are not consistently dominant across all settings. In some cases, the proposed method is not the best-performing approach, so the empirical advantage appears meaningful but not uniformly overwhelming.

Chen et al. Detecting Backdoor Attacks on Deep Neural Networks by Activation Clustering.

Hayase et al., SPECTRE: Defending Against Backdoor Attacks Using Robust Statistics. ICML 2021

---

> ### Author Rebuttal · Authors · 2026-03-29
>
> We thank the reviewer for their thoughtful feedback and highlighting key strengths of our work. We clarify the remaining concerns and questions below.
>
> &nbsp;
>
> ### W1: Similarity to General Outlier Detection in Activations
> We agree with the reviewer that analyzing activations for outlier detection has been explored for other problems. Our work is indeed grounded in outlier detection. However, we argue that our contribution goes beyond applying an existing technique to a new setting. A key message of our paper is that quantifying stealth of a poisoned passage in isolation (e.g., using a standalone classifier on passage text or perplexity filtering) is insufficient, because under limited corruption, a poisoned passage must exert undue influence on the response during inference to be effective. This reframes RAG poisoning defense as a problem of detecting disproportionate influence on the inference process itself, which is a distinct formulation from prior work on backdoor detection in CNNs.
>
> The works cited by the reviewer (Chen et al., Hayase et al.) detect whether a model has been compromised during training by identifying clusters of poisoned samples in the model’s activation space. In our setting, the model itself is not compromised. Instead, we detect whether an input (a single passage) to a clean model has been poisoned by analyzing how the model processes it in the context of the other retrieved passages. Rather than finding a persistent backdoor trigger shared across many inputs, we identify per-query influence anomalies in a single retrieved set at inference time. We observe that our mechanism transfers across models and datasets without heavy retraining or per-configuration tuning (Table 2, 3, 13 and 14). Furthermore, the formalization through SADG and the design of NPAS as a passage-level influence proxy are specific to this problem structure.
>
> We will add a more thorough discussion of prior work on activation-based defenses and clarify these distinctions in the revision.
>
> &nbsp;
>
> ### W2: Comparison to the Baselines
> We agree with the reviewer that Certified Robust RAG (Cert. RRAG) is comparable and better in some configurations. This is primarily due to differences in design goals between the two defenses. Cert. RRAG generates $k$ individual responses using each passage in isolation and robustly aggregates them. It aggressively prunes the context to retain only keywords from individual responses. This strict design results in lower ASRs in some configurations but often at the cost of utility. An inherent limitation is that it cannot handle queries requiring reasoning over multiple passages together. On HotpotQA (Table 14), where multi-passage reasoning is frequently required, AV Filter outperforms Cert.RRAG Keyword in $6/6$ configurations and Cert. RRAG Decoding in $5/6$ in terms of ASR.
>
> Another distinct advantage of AV Filter is its design as a pre-processing step: it does not alter the inference pipeline and has minimal effect on clean accuracy, which means it can be combined with any other defense. In Table 15, we combine AV Filter with Cert. RRAG itself, and AV Filter + Cert. RRAG Keyword achieves higher robust accuracy and lower ASR compared to Cert. RRAG Keyword alone in all $18/18$ settings. This demonstrates the clear benefits of our technique despite not being universally dominant as a standalone defense. We will frame these comparisons more carefully in revision.
>
> &nbsp;
>
> **We again thank the reviewer for recognizing the strengths of our work and for the constructive feedback, which will help us improve the paper. In light of our clarifications, we kindly request the reviewer to consider increasing their rating. We are happy to provide further explanation or any specific experimental evidence on any point.**

---

> > ### Author Rebuttal · Reviewer_CpFQ · 2026-04-06
> >
> > Thanks for the rebuttal, I will keep my score.

---

> > > ### Author Response · Authors · 2026-04-06
> > >
> > > We thank the reviewer for engaging with our rebuttal and for confirming that all concerns have been resolved. We appreciate the positive rating and are glad our responses were helpful. If there is any further suggestion that would help move toward a stronger recommendation, we would be happy to address it.

---

### Official Review · Reviewer_WSjJ · 2026-03-17

**Soundness:** 2
**Presentation:** 3
**Significance:** 2
**Originality:** 3
**Overall Recommendation:** 4
**Confidence:** 2

**Summary:**

The authors propose a method to sanitize the retrieved document set of RAG under poisoning attacks. The idea is to use attention to identify the “outlier” paragraphs that could potentially be poisoned.

**Compliance With Llm Reviewing Policy:**

Affirmed.

**Key Questions For Authors:**

Please see above.

**Limitations:**

Yes

**Strengths And Weaknesses:**

### Presentation

- The paper is overall well-written, the figures and plots are neat.

### Significance

- The topic of the paper is defending against poisoning attacks in RAG systems which is important for the community.
- As a downside, it seems to me that the method is dependent on the document set to contain enough samples with correct answers, so that a potentially poisoned one appears as an outlier.

### Soundness

- The proposed method seems generally effective.
- It would be important to understand how the method works when the retrieved set contains only a small number of relevant documents vs. many relevant documents.
- Another important thing to understand is what happens when the adversary targets a specific question by inserting let’s say multiple similar poisoned samples. Will the method remove the clean documents from the retrieved set because they become the outliers in this case?

### Originality

- I don’t have concerns regarding the paper’s originality.

---

> ### Author Rebuttal · Authors · 2026-03-29
>
> We thank the reviewer for their thoughtful feedback and highlighting key strengths of our work. We clarify the remaining concerns and questions below.
>
> &nbsp;
>
> ### W1: Number of Relevant Documents in Benign Retrievals
> We agree with the reviewer that the number of relevant passages in the benign retrieved set is an important factor for any outlier detection mechanism. Information-theoretically, it is impossible for any generation mechanism to reliably produce a correct answer if the number of poisoned passages equals or exceeds the number of relevant benign passages. In our experiments, we assume that for each corruption fraction $\epsilon$, the number of relevant passages is at least more than $\epsilon \cdot k$. This naturally varies across queries and knowledge bases, which was part of our motivation for evaluating different knowledge sources (Google Search and Wikipedia).
>
> To provide concrete statistics on the number of relevant passages per query, we use Claude Opus 4.6 as an LLM judge. For each query in the clean retrieved set, we prompt the model with the query, the correct answer, and each retrieved passage, obtaining a binary relevance judgement (i.e., whether the passage contains information that helps arrive at the correct answer). The table below reports these statistics across all datasets for the top-10 retrieved passages. Mean and median (out of 10) are expressed in terms of relevant passages per query,  **0 Rel. (%)** is the percentage of queries where none of the retrieved passages were relevant, and **$\geq 1$ Rel. (%)** is the percentage of queries with at least one relevant passage.
>
> | Dataset | Mean | Median | 0 Rel. (%) | ≥1 Rel. (%) |
> |:---:|:---:|:---:|:---:|:---:|
> | Open-NQ | 5.61 ± 3.32 | 6 | 13.0 | 87.0 |
> | Wiki-Open-NQ | 2.46 ± 2.27 | 2 | 18.0 | 82.0 |
> | RealTimeQA | 5.51 ± 3.63 | 6 | 15.0 | 85.0 |
> | Wiki-HotpotQA | 1.92 ± 1.74 | 2 | 18.0 | 82.0 |
>
> Notably, the Wikipedia-based datasets have significantly fewer relevant passages ($1.9-2.5$ on average) compared to Google Search-based datasets ($5.5-5.6$), yet AV Filter still effectively reduces ASR across both settings. For example, on NQ with Mistral-7B ($\alpha = \infty$), AV Filter reduces ASR from $24.6$ to $5.8$ (PIA) and $9.2$ to $4.0$ (Poison) with Google Search (Table 3), and from $55.8$ to $26.8$ (PIA) and $50.4$ to $12.4$ (Poison) with Wikipedia (Table 14). While the absolute ASR is higher on Wikipedia due to fewer relevant passages providing less redundancy, AV Filter still achieves substantial reductions in both cases. This suggests the defense remains robust even when retrieval quality is lower.
> &nbsp;
>
> ### W2: Multiple Poisoned Samples
> This is the flip side of the first concern. It is information-theoretically impossible for any defense to generate a correct answer if the corruption fraction  $\epsilon > 0.5$. We restrict our analysis to $\epsilon < 0.5$ and discuss this in our limitations sections as well (Remark 3.1). Handling majority corruption requires designing better retrieval mechanisms that prevent it from occurring in the first place, potentially using some auxiliary source of trust. We leave this as an important direction for future work on end-to-end robust RAG systems.
>
> In our current work, we evaluate AV Filter under varying corruption levels with $\epsilon = 0.1$ in the main text (all tables by default) and $\epsilon \in \lbrace 0.1, 0.2, 0.3, 0.4 \rbrace$ in Appendix D.8. These results show that AV Filter effectively limits ASR at each level of corruption, with an expected upward trend as the corruption fraction increases. To specifically answer the reviewer’s question: AV Filter would remove benign passages in the scenario where the LLM produces the correct answer despite majority corruption. In this case, the benign passages driving the correct response would exhibit the highest NPAS and be flagged for removal. This is precisely why the $\epsilon < 0.5$ assumption is necessary.
> &nbsp;
>
>
> **We again thank the reviewer for recognizing the strengths of our work and for the constructive feedback. In light of our clarifications, we kindly request the reviewer to consider increasing their rating. We are happy to provide further explanation or any specific experimental evidence on any point.**

---

> > ### Author Rebuttal · Reviewer_WSjJ · 2026-04-01
> >
> > Thank you for the rebuttal. I don't have any further questions.

---

> > > ### Author Response · Authors · 2026-04-04
> > >
> > > We thank the reviewer for engaging with our rebuttal and for confirming that all concerns have been resolved. We appreciate the positive rating and are glad our responses were helpful. If there is any further suggestion that would help move toward a stronger recommendation, we would be happy to address it.

---

### Decision · Program_Chairs · 2026-04-30

**Decision:**

Accept (regular)

**Comment:**

All reviewers agree that the problem is important and timely, and they consistently highlight several strengths of the work. In particular, the paper makes a meaningful conceptual contribution by reframing stealth in RAG poisoning through a distinguishability lens, which several reviewers found insightful. The proposed AV Filter is simple, interpretable, and practical, requiring no retraining and minimal changes to existing pipelines. Empirically, the paper provides a broad evaluation across multiple models, datasets, and attack settings, and demonstrates clear improvements in robustness (e.g., reduced attack success rate while maintaining utility).

**At the same time, reviewers raise a set of recurring concerns:**

1. Validation of attention as a proxy for influence. Multiple reviewers question whether attention-based signals reliably reflect causal influence on model outputs. While the rebuttal provided additional evidence, this remains a conceptual limitation that should be clearly acknowledged and more carefully framed in the final version.

2. Threshold selection and generalization. The AV Filter relies on a threshold estimated from a specific model/dataset setting. Reviewers note that this may limit robustness under distribution shift, and request either stronger justification or more thorough sensitivity analysis.

3. Dependence on retrieval composition and adversarial settings. Questions were raised about how the method behaves when the retrieved set has few relevant documents, or when multiple similar poisoned passages are inserted, potentially altering outlier structure.

4. Novelty relative to prior work. One reviewer points out that detecting adversarial or poisoned samples via outlier signals in internal representations has precedents (e.g., activation clustering and robust statistics methods), and views this work as an extension of that line into the RAG setting.

5. Evaluation limitations. Concerns include limited scale of adaptive attack experiments and questions about fairness of some baseline comparisons.

After the rebuttal, all reviewers indicated that their concerns were sufficiently addressed and maintained positive ratings.

Overall, I find that the paper makes a solid and well-executed contribution. While some aspects (e.g., reliance on attention signals and thresholding) are not fully resolved and slightly weaken the conceptual grounding, the work is technically sound, practically relevant, and empirically thorough. Given the uniformly positive reviewer feedback after rebuttal, the importance of the problem, and the practical value of the proposed approach, I recommend Weak Accept. The paper would benefit from clearer discussion of its assumptions and limitations.